# Detection of land surface induced atmospheric water vapor patterns

Tobias Marke[1], Ulrich Löhnert[1], Vera Schemann[1], Jan H. Schween[1], and Susanne Crewell[1]

[1]Institute for Geophysics and Meteorology, University of Cologne, Germany

**Correspondence:** Tobias Marke tmarke@meteo.uni-koeln.de

**Abstract.** Finding observational evidence of land surface atmosphere interactions is crucial for understanding the spatial and temporal evolution of the boundary layer, as well as for model evaluation, in particular large-eddy simulation (LES) models. In this study, the influence of a heterogeneous land surface on the spatial distribution of atmospheric water vapor is assessed. Ground-based remote sensing measurements of a scanning microwave radiometer (MWR) are used in a long-term study over six years to characterize spatial heterogeneities in integrated water vapor (IWV) during clear sky conditions at the Jülich Observatory for Cloud Evolution (JOYCE). The resulting deviations from the mean of the scans reveal a season- and direction-dependent IWV that is visible throughout the day. Comparisons to a satellite derived spatial IWV distribution show good agreement for a selection of satellite overpasses during convective situations, but no clear seasonal signal. With the help of a land use type classification and information on the topography, the main type for the regions with a positive IWV deviation was determined to be agricultural fields and nearby open pit mines. Negative deviations occurred mainly above elevated forests and urban areas. In addition, high resolution large-eddy simulations (LES) are used to investigate changes in the water vapor and cloud fields for an altered land use input.

## 1 Introduction

Interactions between the land surface and the atmospheric boundary layer can have significant influences on the regional weather and climate. Heterogeneity in land use, among other parameters characterized by soil type, vegetation and urban areas, induces spatial variability in surface fluxes of momentum, sensible and latent heat. Numerical studies suggest, that contrasts in land surface fluxes are responsible for mesoscale circulations and considerably affect the state of the atmospheric boundary-layer in a non-linear way (e.g. Ookouchi et al., 1984; Pielke et al., 1991; Clark and Arritt, 1995). On a more local scale the transport of energy and water vapor into the atmosphere can trigger the formation of shallow convective clouds and precipitation (e.g. Rabin et al., 1990; Avissar and Schmidt, 1998). Because this small scale variability can not be resolved by most weather forecast and climate models, it needs to be parameterized. This requires assumptions near the surface boundaries, which strongly affects exchange processes. Unresolved patterns in the models are crucial, since the resulting gradients directly influence the fluxes and hence the evolution of the model state (Simmer et al., 2015). Monitoring and modeling these spatial

patterns and interactions is the main focus of this study, which is conducted within the framework of the Transregional Collaborative Research Centre 32 (TR32) "Patterns in Soil-Vegetation-Atmosphere Systems" (www.tr32.de). The scope of TR32, as described in Simmer et al. (2015), is to improve the understanding and prediction capabilities of the spatiotemporal evolution of the terrestrial system across scales using measurement techniques and modeling platforms by integrating activities of several research groups.

Since the scales of surface heterogeneity and resulting interaction processes with the overlying boundary-layer are on the order of meters to kilometers, a frequently used tool for studying these interaction processes on a local scale is conducting high resolution large-eddy simulations (LES) (e.g. Courault et al., 2007; Huang and Margulis, 2009; Maronga and Raasch, 2013; Shao et al., 2013). By altering the land surface properties, the turbulence resolving simulations provide estimates of the resulting effect on the boundary-layer structure. In this way Vilà-Guerau De Arellano et al. (2014) show differences in cloud dynamics that can be related to the partitioning of the surface fluxes determined by the plant functional type. In van Heerwaarden and Vilà-Guerau De Arellano (2008) an enhancement of cloud formation over heterogeneous landscapes using different Bowen ratios is indicated.

For a better understanding of the influence of the land surface on the atmospheric state, and in order to evaluate model findings, ground-based observations by current state-of-the-art remote sensing instrumentation can be used. Significant effects of heterogeneous land use on the turbulent fluxes and connections to clouds have been shown in several field campaigns in a short-term perspective (Weckwerth et al., 2004; Beyrich et al., 2006; Wulfmeyer et al., 2011; Späth et al., 2016; Macke et al., 2017; Wulfmeyer et al., 2018). Investigating the influence of land use heterogeneity on boundary-layer characteristics, such as water vapor and clouds from long-term measurements can play a key role in finding systematically significant patterns in relationships between the local land surface and atmosphere above.

As a key parameter that connects vegetation activity and the boundary-layer, atmospheric water vapor plays an important role within the hydrological cycle, but also for the energy balance at the surface and within the atmosphere. Späth et al. (2016) investigated water vapor fields for a limited amount of time in a campaign with a scanning differential absorption lidar and found gradients related to surface elevation and land cover type. But also long-term studies of the spatiotemporal variability of water vapor have revealed terrain-related processes in a mountainous area (Adler et al., 2016) by using scans of a passive ground-based microwave radiometer (MWR). Compared to the widely used satellite observations for spatially resolved water vapor estimates, available only for a handful of overpasses per day, the MWR is well suited for continuous and temporally highly resolved measurements at a certain location. While MWR profile measurements of humidity suffer from coarse resolution, a good agreement between zenith measurements of integrated water vapor (IWV) using also MWR, satellite and Global Positioning System (GPS) observations was shown in Steinke et al. (2015). The MWR has already proven to be able to detect horizontal humidity gradients by retrieving IWV values in a scanning configuration (Kneifel et al., 2009; Schween et al., 2011).

To address the question whether spatial water vapor distributions can be connected to land surface properties, this observational and modeling study focuses on the long-term pattern of azimuthal IWV deviations derived from satellite and ground-based measurements at the Jülich ObservatorY of Cloud Evolution (JOYCE, Löhnert et al. (2015)) in Western Germany (50.91°N, 6.41°E). At JOYCE, various remote sensing instruments, including a scanning MWR, have been deployed since 2011 to continuously monitor water vapor, clouds and precipitation. For comparing the spatial IWV distribution derived from the MWR with an independent measurement, a satellite water vapor product is used at high spatial resolution. In addition, a Doppler wind lidar is available for a characterization of the atmospheric boundary-layer in terms of the winds and turbulent mixing processes that control the exchange of water vapor between the surface and the atmosphere. The impact of the land surface on the atmospheric water vapor distribution is evaluated by comparing the derived IWV deviations to a detailed land use map. To better understand the impact of the land surface on the evolution of the cloudy boundary-layer, sensitivity studies with high resolution LES are performed with different land use type settings.

The details of the instruments and data used in this study in Sect. 2 is followed by the description of the data sample derivation used in the long-term analysis. For a better description of the state of the boundary-layer during clear-sky conditions and large scale effects, the results are shown together with wind and turbulence statistics derived from Doppler lidar measurements during the MWR scans and a reanalysis product (Sect. 3.1). Subsequently, the IWV deviations derived from MWR scans and for a collection of satellite overpasses are compared for different seasons (Sect. 3.2) and for a selected single day. A model case study is complemented by the analysis of two large-eddy simulations focusing on the land use influence on the evolution of the cloudy boundary-layer (Sect. 4) and a summary of the results is given in Sect. 5.

## 2 Instruments and data

### 2.1 Microwave radiometer

The microwave radiometer HATPRO (Humidity And Temperature PROfiler) at JOYCE utilizes direct detection receivers and measures the brightness temperatures (TB) at 7 channels in the K-band from 22 GHz to 32 GHz and at 7 channels also in the V-band from 52 GHz to 58 GHz. In this study, the observations of the 7 K-band channels with a 1–2 s temporal resolution are taken into account. A statistical approach based on a least squares linear regression model (Löhnert and Crewell, 2003) is applied to derive IWV, absolute humidity (q) and liquid water path (LWP) using observations of the downwelling microwave radiance along the water vapor absorption line between 22.24 and 27.84 GHz and in the atmospheric window at 31.4 GHz. The instrument is capable of high temporal resolution (Rose et al., 2005) and the absolute error in zenith TB measurements of 0.5 K is mainly determined by the instrument absolute calibration (Maschwitz et al., 2013). This accuracy converts into an uncertainty of 0.5–0.8 $\text{kg m}^{-2}$ in the derived IWV and 20–30 $\text{g m}^{-2}$ for LWP.

The zenith measurements ($\text{IWV}_z$) alternate with full azimuth scans in 10° steps at 30° elevation angle. The degrees of freedom for signal (DFS) are usually between 1–2 for MWR humidity retrievals and the highest information content can be found

in the boundary-layer. For the zenith retrieval 1.87 DFS and for the 30° (slant path) retrieval 2.14 DFS are identified. The scans are available between June 2012–June 2015 and starting from June 2018. In 2016 and 2017 no MWR scans were performed. The scanning frequency is 15 min and is increased to 10 min between 25 June and 18 July 2018 and decreased to 30 min after 18 July 2018. Due to directional dependent interference in the unprotected 26.24 GHz channel, specific azimuth directions are not considered (50°, 160°, 180°, 260°). Since the excluded azimuth directions are not connected, no larger gap is apparent and a smooth transitions between the gaps can be assumed. Therefore the missing IWV values are filled using linear interpolation. For all scans, the derived LWP, IWV and q are air-mass corrected to account for the slant angle of the scanning MWR.

## 2.2 Doppler lidar and boundary-layer classification

As a pulsed lidar system, the Halo Photonics Streamline Doppler lidar (Pearson et al., 2009) provides range-resolved profile measurements of radial Doppler velocity and backscattered signal. With a wavelength of 1.5 $\mu$m (near-IR) the instrument is sensitive to the backscatter of aerosols and clouds and is able to scan the full hemisphere. The maximum detectable range depends on the presence of atmospheric particles and the lowest reliable range is at 105 m. At JOYCE the system is set to a range resolution of 30 m and performs plan position indicator scans every 15 min to estimate wind speed and direction profiles based on the velocity-azimuth display (VAD) method using 36 beams at 75° elevation. In addition the Doppler beam swing (DBS) technique with three beams and range height indicator scans are scheduled every 5 min and 30 min, respectively. For the remaining time, the instrument is staring at zenith to derive the vertical velocity with high temporal resolution (1 s).

To study land surface atmosphere exchange processes it is crucial to know the turbulent state of the boundary-layer. Therefore an objective classification of the mixing sources presented by Manninen et al. (2018) is utilized to describe the turbulence characteristics during MWR scans at JOYCE. The method is based on the combination of multiple Doppler lidar quantities including the dissipation rate of turbulent kinetic energy (TKE) derived from vertically pointing observations using the method presented in O'Connor et al. (2010). The TKE dissipation rate is based on the variance of the observed mean Doppler velocity and allows for a threshold based estimation of the convective boundary-layer (CBL) height by determining the last range bin in each profile with significant turbulence in a bottom-up approach.

## 2.3 MODIS IWV

The passive, imaging Moderate Resolution Imaging Spectroradiometer (MODIS) measures in 36 spectral bands ranging from 0.4 $\mu$m to 14.4 $\mu$m. Two MODIS instruments are currently airborne on NASA's sun-synchronous near-polar-orbiting Earth Observing System Terra and Aqua satellites. A full coverage of the globe is achieved in 1–2 days with an orbit height of 705 km and a scan rate of 20.3 rpm. The swath dimension of MODIS is 2330 km (cross track) and 10 km (along track at nadir). Within the 36 spectral bands, five channels in the 0.8–1.3 $\mu$m near-infrared spectral region can be used for water vapor remote sensing (Gao and Kaufman, 2003). For IWV estimates the Level-2 (Collection 6.1) near-infrared retrieval (MODIS-NIR) with a 1 km spatial resolution is chosen. The retrieval by Gao and Kaufman (2003) is based on three channels at 0.936 $\mu$m, 0.940 $\mu$m

and 0.905 $\mu$m for the water vapor absorption and at 0.865 $\mu$m and 1.24 $\mu$m to correct for atmospheric gaseous absorption. In order to derive the total vertical amount of water vapor, the reflected NIR solar radiation in the water vapor absorption channel is compared to the window channels yielding the atmospheric water vapor transmittance. The amount of water vapor is then obtained from look-up tables derived from a line-by-line atmospheric transmittance code. Reliable estimates of the water vapor total column amount over land areas can only be inferred during daytime and for cloud free regions. Typical errors of the MODIS-NIR water vapor product range between 5–10%. Here, a height correction similar to Steinke et al. (2015) of the retrieved values is performed due to the variations of the horizontal and height distance to JOYCE per flight track of MODIS. The height difference is corrected by assuming an exponential decrease of the humidity profile and by using the water vapor density obtained from measurements of temperature, humidity and pressure of a weather sensor attached to the MWR and the topography with a 200 m horizontal resolution. Furthermore, the IWV product was resampled to 100 m for calculating the mean values of several overpasses.

## 2.4 ERA5 data products

To distinguish between local influences and large scale features regarding the observed spatial pattern of IWV deviations, the reanalysis products of ERA5 with a 31 km horizontal resolution are analyzed (Copernicus Climate Change Service (C3S), 2017). Besides the u and v wind components at different pressure levels (1000 hPa, 700 hPa), the direction of the IWV transport (IWVT, in degrees) is also considered at a 3 h temporal resolution for the closest point to JOYCE. The vertical integral of water vapor flux, used to derive IWVT, is calculated utilizing the specific humidity and winds on model levels. The ERA5 IWV is selected at the closest output time to the MWR scans.

## 2.5 ICON-LEM

As a state-of-the-art atmospheric modeling system, the ICOsahedral Non-hydrostatic model ICON (Zängl et al., 2015) has been developed by the German Weather Service (DWD) and the Max Planck Institute for Meteorology (MPI-M). The ICON Large-Eddy Model (ICON-LEM) was designed within the framework of the High Definition Clouds and Precipitation for advancing Climate Prediction (HD(CP)[2]) project for improving moist processes in climate prediction models (Heinze et al., 2017). In this study, the ICON-LEM simulations are used to provide a spatial representation of the IWV field to compare with the measurements obtained from the scanning MWR and the MODIS-NIR water vapor product around JOYCE.

A good agreement between simulations of ICON-LEM using high grid resolutions of up to 156 m and observations was already shown in Heinze et al. (2017) concerning turbulence, column water vapor, and cumulus clouds (compared to satellite observations). The topographic influence on the wind field was also shown in ICON-LEM simulations and observations at JOYCE (Marke et al., 2018). Therefore a similar setup with a domain radius size of 10 km, 78 m horizontal resolution and 20 km vertical extent is used in this study. The minimal layer thickness is 20 m and the lowest 2 km contain 33 levels. Initial and lateral boundary conditions are created from the output of the ECMWF Integrated Forecasting System (IFS) model. As

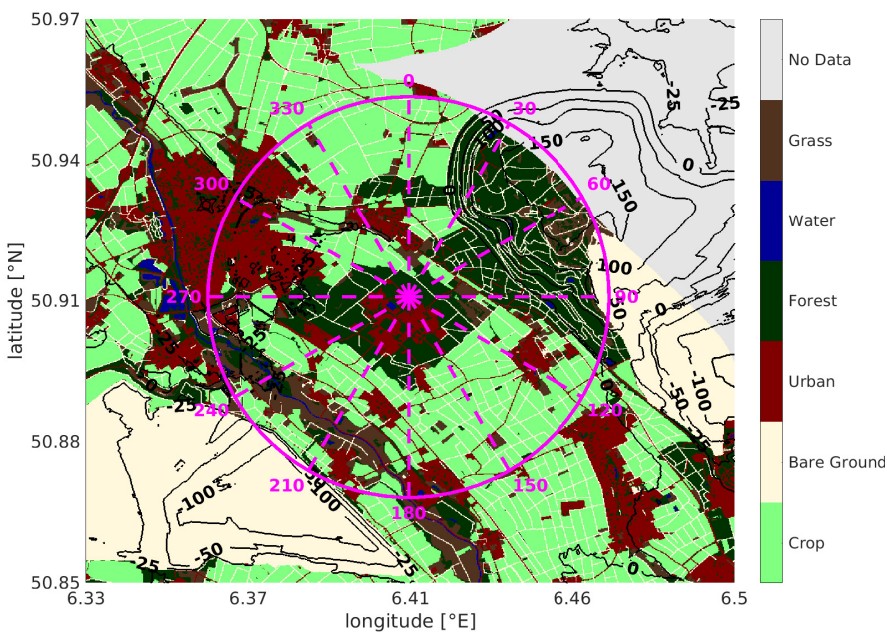

**Figure 1.** Simplified map (12x13 km) of the land use classification described in Waldhoff et al. (2017) centered around JOYCE. The circle (4.3 km radius) shows the crossing distance and azimuth angles of the MWR scans at the IWV scaling height of 2.5 km. Contours refer to the height relative to JOYCE (111 m a.s.l.).

the IFS and the ICON model do not use an identical land surface model, a sensitivity of the simulations to the treatment of soil moisture and other land surface components can not be excluded. But those sensitivities are the same for both simulations and sensitivity studies implicate, that the results are rather robust despite small variations. In addition to the control simulation using a simplified version of the land use input data GLOBCOVER (Bontemps et al., 2011) with 300 m resolution, a second
5    simulation is conducted with one altered land use setting. In this way parameters like leaf area index and roughness length are changed to get a different distribution of potential water vapor sources and sinks at the surface.

## 2.6   Land use classification and measurement site description

To be able to link atmospheric water vapor measurements to land surface properties, spatial land use information is needed. This is addressed by using a remote sensing-based regional crop map (Waldhoff et al., 2017) that was applied to a study area
10   in Western Germany including the surrounding area of JOYCE. In this method, supervised multi-temporal remote sensing data of Sentinel-2, ancillary information and expert-knowledge on crops are combined in a Multi-Data Approach (MDA). The classification is therefore able to differentiate between 44 vegetated, urban and water areas with a spatial resolution of 15 m.

The detailed and highly resolved classification is used to identify areas with a predominant land use type. Therefore the classified types are condensed into six main types, in particular agricultural areas, grassland, bare ground, urban areas, deciduous forest and water. These six groups are expected to have a significantly different behavior in terms of transpiration and/or evaporation depending on the season and therefore might cause atmospheric water vapor patterns that can be distinguished and related to the appropriate type. In Fig. 1 the simplified land use classification of a 12x13 km area centered around JOYCE is shown. The city of Jülich to the northwest but also JOYCE at the Research Center Jülich are the largest urban areas in this surrounding. The artificially created pit mine dump hill Sophienhöhe is located in the northeast direction, which is up to 200 m higher than JOYCE and covered mainly by a deciduous forest. In the northern and southeastern part of the selected domain mostly agricultural sites can be identified. The main crop type between April and June is winter wheat, and sugar beet, maize and potato are dominant from July until September. A common crop rotation is a two year cycle of sugar beet to winter wheat (Waldhoff et al., 2017). Due to this crop rotation and regarding the small field sizes in this domain, no further distinction in crop types is made, but more active crop fields in terms of evapotranspiration are present during the spring season. The southwestern parts are mostly crop fields but also grasslands surrounding the Rur River, with its valley going from southeast to northwest. The pit mines (bare ground) with depressions down to 300 m below JOYCE are located to the east and southwest.

## 3 Long-term observed directional IWV deviations

### 3.1 Data sample derivation and characteristics

In order to find patterns in the long-term water vapor scans at JOYCE, that can be related to local land surface characteristics, the MWR scans are evaluated during meteorological conditions that are favorable for strong land surface atmosphere interactions. This excludes overcast situations and large scale advection of moist or dry air as during these the surface influence is low as shown by Steinke et al. (2019) by the amplitude reduction of the diurnal water vapor cycle. The cloud detection is obtained by using the 31.4 GHz channel, which is within an atmospheric window. The signal from this channel is dominated by the presence of liquid water in the case of clouds appearing in the instrument's field of view. During a single scan the maximum difference of the measured 31.4 GHz brightness temperature for each azimuth direction and the mean of the whole scan must be below 2 K, since liquid water clouds are expected to cause a much higher difference. Furthermore the air-mass corrected LWP from the statistical retrieval needs to be below $20\,\mathrm{g\,m^{-2}}$, which is on the order of the retrieval uncertainty. To avoid scenes with large scale advection of moist or dry air, the difference between the maximum and minimum $\mathrm{IWV_z}$ within one hour around the scan needs to be smaller than $2\,\mathrm{kg\,m^{-2}}$. This threshold is chosen to be above the instrument sensitivity for IWV. These requirements need to be fulfilled for at least three consecutive scans. The first and last scan of each sequence are neglected to ensure that they are not part of a transition from conditions violating the criteria. The choice of the thresholds showed to be a good trade-off between excluding apparent cloudy situations, but still allowing a sufficient number of scans to generate a large data sample. In order to detect seasonal differences due to different stages of crop development from the growing phase over senescence to harvest, the months between April–June and July–September between 2012–2018 are separated. The highest diurnal IWV variability is observed between spring and autumn at JOYCE (Löhnert et al., 2015) and the influence from the land

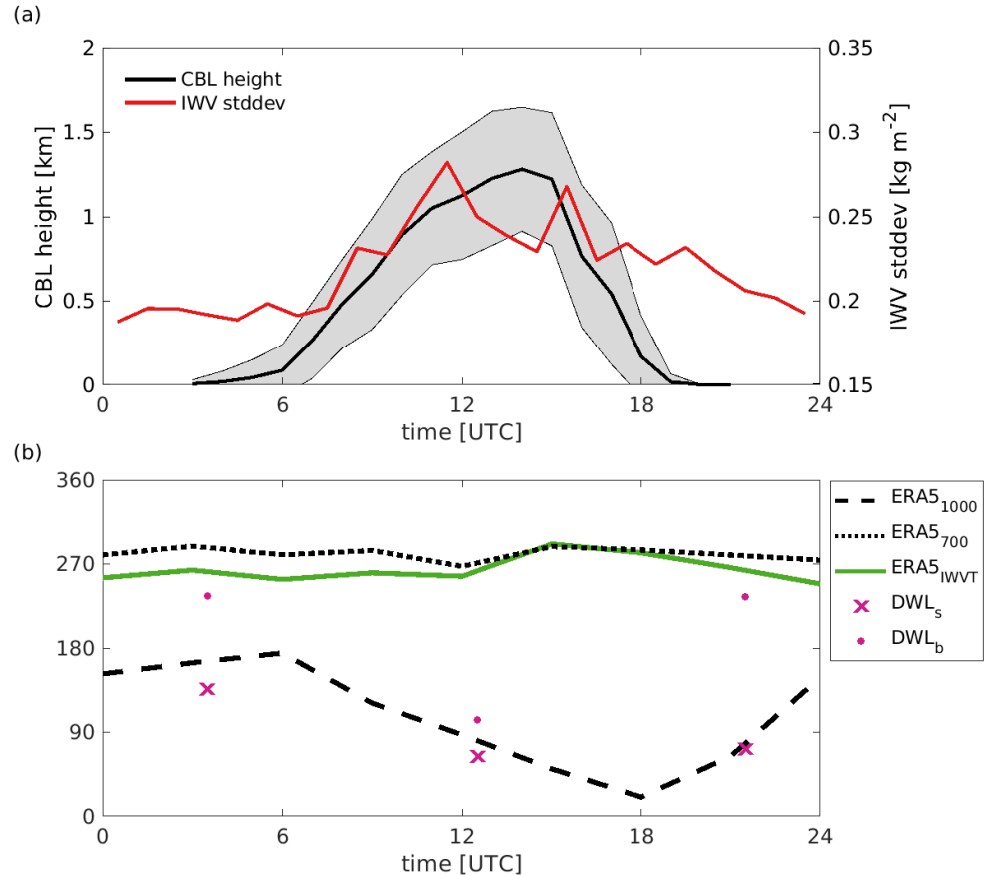

**Figure 2.** (a) Hourly averaged convective boundary-layer (CBL) height (with standard deviation in shadings) from the Doppler lidar boundary-layer classification at the MWR scan times. The zenith IWV standard deviation (stddev) is determined within 1 h around the scans. (b) The lines show the directions (in degree) of the averaged ERA5 wind directions at 1000 hPa ($ERA5_{1000}$), 700 hPa ($ERA5_{700}$) and the IWV transport ($ERA5_{IWVT}$). Symbols indicate the mean Doppler lidar wind direction (average times: 01–06 UTC, 10–15 UTC, 19–24 UTC) at 105 m ($DWL_s$) and 1005 m ($DWL_b$).

surface is expected to be largest in spring and early summer. Instead of using the total slant column IWV, the humidity profile is integrated up to the CBL height determined by the Doppler lidar (hereafter: $IWV_{CBL}$) for an analysis of the lower tropospheric water vapor patterns. For all scans, the mean value per scan is subtracted to investigate the deviations in each azimuth direction.

5    In addition, a co-located Doppler lidar is used to gain information on atmospheric turbulence, wind direction and wind speed during the scans. The temporal resolution of the Doppler lidar VAD scans is 15 min and the closest measurement to the scan time is selected. For the general development of the wind direction during the day, 6 h averages are calculated. The

number of MWR scans per hourly bin that meet the requirements ranges from 127 to 496 with fewer scans during midday. The decrease in number of cases during daytime is due to the formation of convective clouds, since overcast situations would influence the number of cases independent of the time of the day. The mean standard deviation for each scan increases from 1.1% to 1.94% during daytime indicating the influence of convective activity, which is shown by high TKE dissipation rates and a corresponding mean CBL height up to 1.28 km (Fig. 2(a)). Also the IWV standard deviation from the zenith MWR measurements in Fig. 2(a) reveals a diurnal cycle during this measurement period of late spring until early autumn, which is in agreement with the seasonal statistics derived in Löhnert et al. (2015). While the IWV standard deviation follows the rate of the CBL height development in the morning hours, an abrupt decrease is only evident in the turbulence measurements in the afternoon transition period. This suggests that water vapor is mixed into the upper layers of the atmosphere during daytime and is still present in the residual layer throughout the night.

For assessing the impact of the large scale water vapor transport, the ERA5 reanalysis product is used. The ERA5 IWV at the closest output time to the MWR scans compared to the 1 h averaged $IWV_z$ from the MWR shows a high correlation coefficient of 0.98 and a root-mean-square error (RMSE) of only 1.46 $kg\,m^{-2}$. The ERA5 wind direction at 1000 hPa ($ERA5_{1000}$) is in good agreement with the mean near surface wind direction (average times: 01–06 UTC, 10–15 UTC, 19–24 UTC) derived from the Doppler lidar at 105 m ($DWL_s$, Fig. 2(b)). The wind direction ranges from a southerly flow during night to an east to north direction during the day corresponding to fair weather situations and anticyclonic flow at this site. The wind direction turns clockwise with height for the ERA5 product and the Doppler lidar observations, but stays relatively constant within the CBL as there is no large difference between $DWL_s$ and the Doppler lidar wind direction at 1005 m ($DWL_b$) between 10–15 UTC. The wind direction in the free troposphere at 700 hPa shows no significant diurnal cycle. The same applies to the IWVT, which corresponds to the westerly wind direction at 700 hPa, showing the west-wind-zone transport of humid air at the mid-latitudes. But at midday and early afternoon (12–17 UTC) positive $IWV_{CBL}$ deviations in the long-term MWR scans increase and shift to the southeast (not shown). Despite the fact, that the ERA5 IWV shows a diurnal cycle, this shift can not be seen in the IWVT, suggesting that also local influences contribute to the observed IWV signal. This is further analyzed in section 3.2.

Separating all cases according to the low-level wind direction from the Doppler lidar, a directional dependence is found related to the wind speed, indicating local transport and a shift between the humidity field and the underlying surface within the MWR scanning beam. To exclude this process and to better connect the spatial IWV deviations with the surrounding land use, the MWR scan analysis is restricted to cases with wind speeds below the median value of 5 $m\,s^{-1}$. During the observational period, 161 days out of a total of 1242 single scans are selected with a mean $IWV_z$ of $18.02 \pm 6.43$ $kg\,m^{-2}$ measured in a 1 h window around the scans. At JOYCE the average year-to-year variability in terms of humidity is rather small, but still a good coverage of relatively dry and wet years is achieved in this study. As an exemplary measure, the mean zenith IWV taken around the selected scans for each year ranges from 15.0 $kg\,m^{-2}$ to 21.4 $kg\,m^{-2}$. Therefore the variability of the zenith IWV values for the different years (4.2-7.8 $kg\,m^{-2}$) is in the range or higher than changes in the mean value.

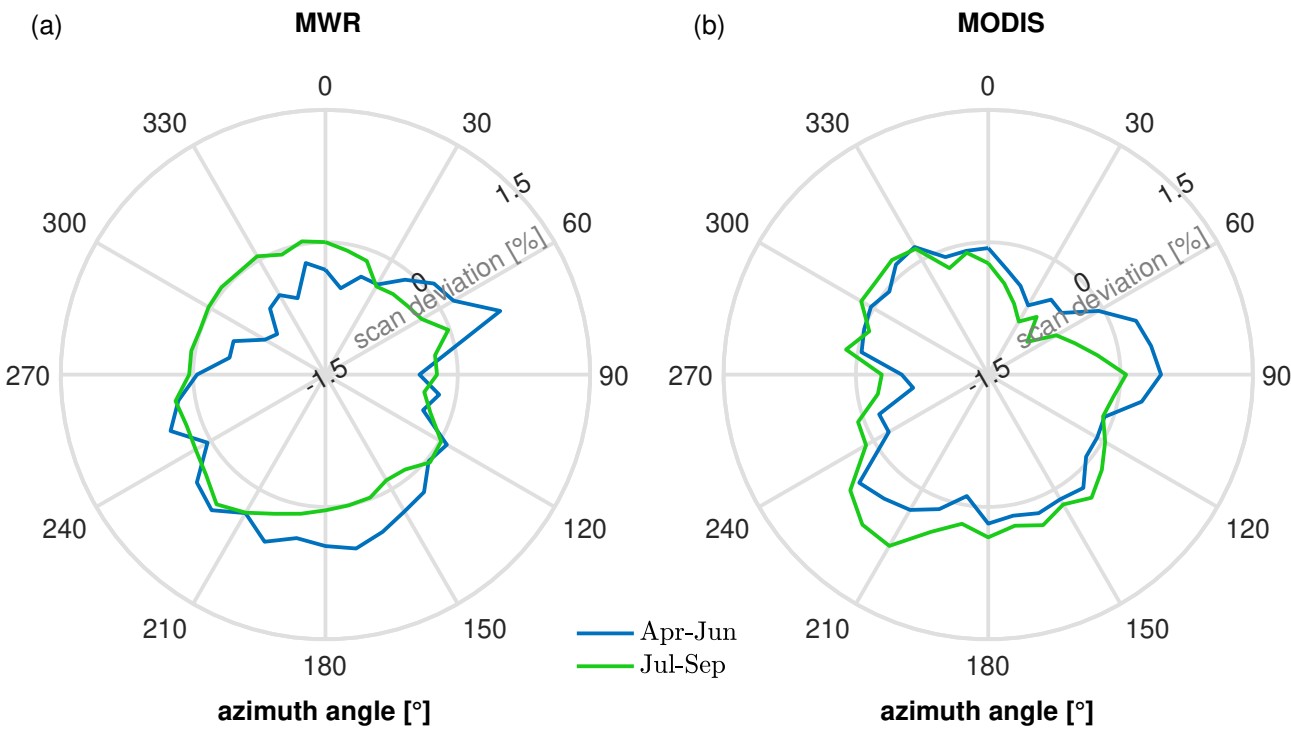

**Figure 3.** (a) Mean values of the MWR water vapor deviation (integrated up to the CBL height) from the mean per scan between 12–17 UTC for April–June (99 scans) and July–September (123 scans). (b) Same as (a) but for the MODIS IWV deviations including 22 overpasses (April–June) and 36 overpasses (July–September), respectively.

### 3.2 Daytime MWR and MODIS derived IWV deviations and connection to land use

Figure 3(a) shows the daytime (12–17 UTC) mean value of the $\mathrm{IWV_{CBL}}$ deviation for all 36 azimuth directions of the MWR scans. In this time period a well-mixed CBL has developed and the highest convective water vapor flux from the land surface into the atmosphere is expected. For the April–June cases a positive deviation up to 0.61% from the mean between 130°–270°

5 is visible. Also a positive peak around 75° is present. Whereas between 270°–60° mostly negative $\mathrm{IWV_{CBL}}$ deviations are present (up to -0.79%). In contrast, the July–September cases only show a positive deviation between 180°–270° and slightly negative between 0°–120°. Otherwise there is no noticeable deviation during this season. Note that these deviations are median values to detect the long-term pattern and that single scan deviations from the mean can reach over 5%.

10 For a comparison with an independent IWV measurement and to exclude that the patterns are influenced by interference, the MWR results are compared to the MODIS-NIR derived IWV around JOYCE. The findings presented here could also be valuable for further studies using the MODIS products for assessing spatial IWV differences, which is especially valuable for larger areas. For a fair comparison of the column amount of water vapor from MODIS to the path-integrated water vapor

observations from the MWR scans, a virtual MWR scan is derived from the MODIS observations. Therefore the total IWV is distributed to an absolute humidity profile for each MODIS pixel assuming a linear decrease by 20% in the CBL and an exponential decrease above, similar to Schween et al. (2011). The mean CBL height is determined from the Doppler lidar based boundary-layer classification (Manninen et al., 2018) around 1 h of each overpass. The CBL height is assumed to be

constant in the area of interest, as well as the $1/e$ height for the exponential decrease, which is calculated from the MWR humidity profile of the corresponding overpass. In this way a virtual scan corresponding to the MWR scan configuration can be performed around JOYCE where the amount of water vapor is integrated for each beam up to the CBL height. Only overpasses without missing data due to the MODIS quality checks are considered.

As an additional comparison of MWR and MODIS, the $\text{IWV}_\text{z}$ measurements of the MWR ($\text{IWV}_\text{z,MWR}$) and the MODIS mean total column amount 1 km around JOYCE ($\text{IWV}_\text{z,MODIS}$) are compared. The zenith IWV values are highly correlated (0.96) with a RMSE of $2.45\,\text{kg}\,\text{m}^{-2}$, which is about $1\,\text{kg}\,\text{m}^{-2}$ higher than found in Steinke et al. (2015). This discrepancy is probably caused by a greater IWV variability shown in Fig. 2(a). For larger IWV values, the MODIS observations tend to an overestimation. For the 22 (April–June) and 36 (July–September) MODIS overpasses occurring between 9–13 UTC, the mean

IWV deviation from the virtual scans are calculated (Fig. 3(b)). Note that only showing the MWR scans during the MODIS overpasses does not change the deviation pattern significantly. In general, the relative deviations from the MODIS virtual scans do not show a seasonal pattern as for the MWR scans (Fig. 3(b)). With both observations, a noticeable negative deviation around 30° is visible, but the agreement in the location of the positive deviations for both seasons around 180°–240° is also evident. This area shows a high fraction of grassland, the Rur River and one of the pit mines explaining the positive deviations

in both seasons whereas less water vapor seems to be present in the vicinity of the forested hill (Fig. 1). Regarding the MODIS derived results, the pit mine around 90° also reveals a positive deviation, but the peak for the MWR is shifted to 70°. This phenomena might be explained by the orographic flow which is strongly altered by the pit mines as shown in Marke et al. (2018) and the low spatial resolution of the MODIS IWV product.

The results presented here for the MWR and MODIS scans suggest a higher water vapor flux into the atmosphere for the agricultural fields in the southwest due to evapotranspiration (no irrigation) especially in the main crop growing season between April–June. The high amount of water vapor around the pit mines could be caused by irrigation to reduce dust emissions during the day and dew formation at night. In contrast, the forest and urban areas reveal a lower water vapor amount. This can be explained by less water availability in urban areas and a higher water use efficiency for deciduous forests compared

to crop fields demonstrated in Tang et al. (2015). A similar difference in the surface fluxes between crops during the main vegetation period and forest (pine trees) was found using surface flux measurements (Beyrich et al., 2006) and in the LES study by Garcia-Carreras et al. (2011). In addition, lower wind speeds due to the topography and a higher roughness length at the forested hill can cause decreased water vapor fluxes into the atmosphere. Thus, spatial water vapor differences can be detected by the scanning MWR, especially in a long-term perspective using a composite of carefully selected cases.


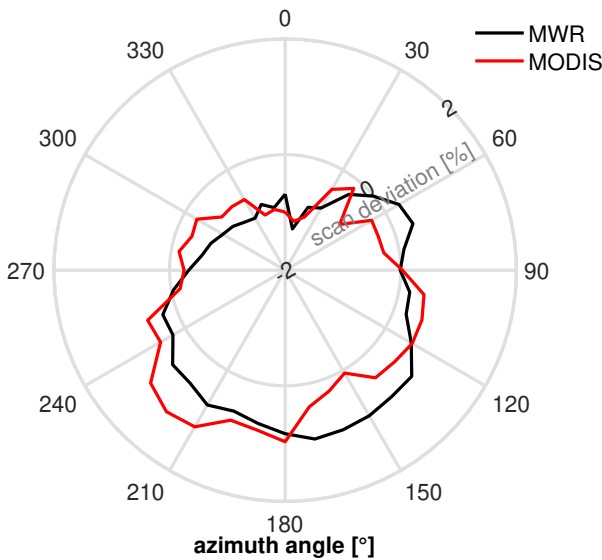

**Figure 4.** Mean values of the MWR and MODIS water vapor deviation for the 25 July 2012 case study including five MWR scans (9:10–11:10 UTC) and one MODIS overpass at 10:15 UTC.

## 4  LES case study analysis for land surface impact

The influence of the land use type on the evolution of the cloudy boundary-layer is further investigated in a case study (25 July 2012) by means of a large-eddy simulation using the ICON-LEM model. Due to the spatial resolution of the land use data and the scale of the land use patches around JOYCE, the crop and grass types are combined. On this day, with a northwesterly wind direction, no clouds are present until 11:30 UTC. The timing of the selected MODIS overpass is 10:15 UTC and five MWR scans are performed between 9:10–11:10 UTC. The results of the observed water vapor deviations are shown in Fig. 4. As already shown in the previous long-term analysis, the maximum positive deviation occurs in a southeasterly to southwesterly direction with a good agreement in the sign changes between MWR and MODIS. Although this day is in late July, it still shows similar features compared to the April–June season, suggesting still active crop fields (especially sugar beet) in this area. In order to make a general statement whether the ICON-LEM is correctly representing the spatial water vapor distribution, a large number of high resolution simulations would be needed. Here, the focus is on assessing the impact of different land use data as input for the model on boundary-layer development and cloud formation. In this 2 hour time interval the CBL height determined by the Doppler lidar increases from 405 m to 1275 m. In the first ICON-LEM simulation (ICON1) using the simplified GLOBCOVER land use data (Fig. 5(a)), the model boundary-layer height reaches these heights about one hour later than in the observations. The mean $IWV_z$ values are 24.83 $\mathrm{kg\,m^{-2}}$ (MWR), 29.26 $\mathrm{kg\,m^{-2}}$ (MODIS) and 28.22 $\mathrm{kg\,m^{-2}}$ (ICON1), where the ICON1 zenith IWV is averaged within a radius of 1 km around JOYCE and for MODIS the nearest pixel is chosen. The lower observed IWV value by the MWR and higher CBL height compared to ICON1 suggests that the partitioning of surface

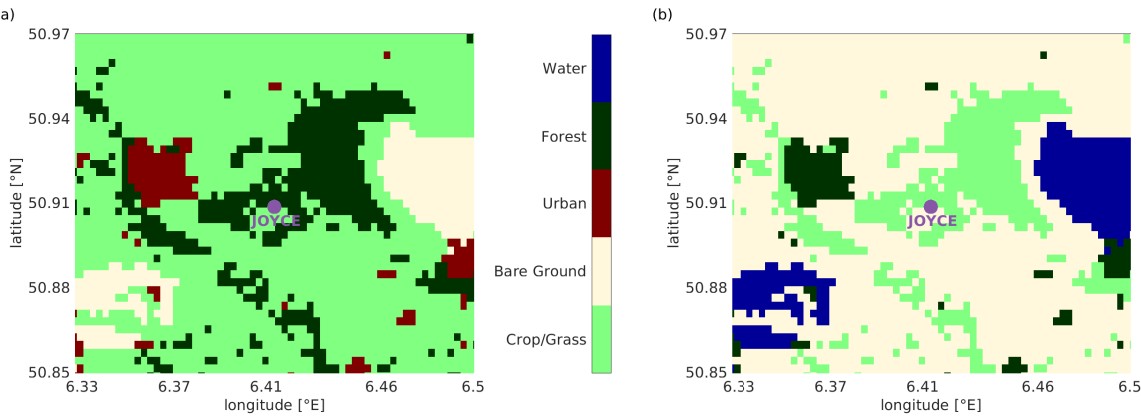

**Figure 5.** (a) 12x13 km map of the simplified GLOBCOVER land use data centered around JOYCE used for the first ICON-LEM simulation (ICON1). (b) Same as (a) but with altered land use types for the second simulation (ICON2).

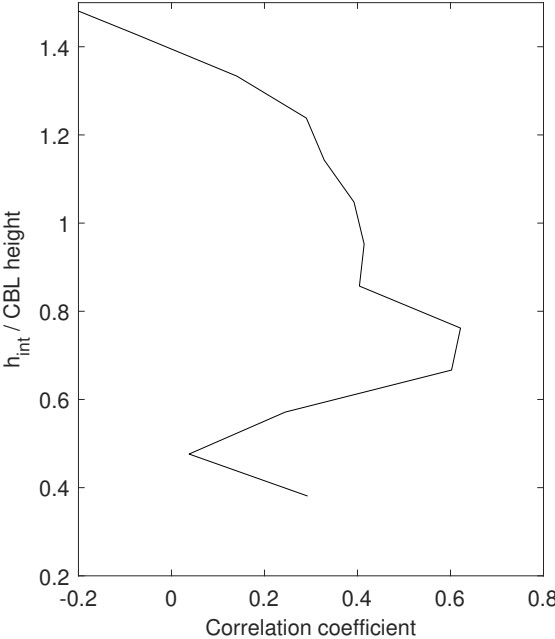

**Figure 6.** Correlation between $10°$ sector estimates of term II of Eq. (2) and the slant path integrated water vapor at $30°$ elevation and $10°$ azimuth steps. $h_{int}$ is the maximum height of the slant path that is used for the integration and is normalized by the CBL height.

heat fluxes is more towards the latent heat flux in the simulation.

Using the humidity budget equation, the contribution of local and non-local sources to the change in atmospheric humidity within the boundary-layer can be estimated using ICON-LEM. The Reynolds averaged continuity equation with contributions only from advection and turbulent flux divergence (no molecular diffusion or other source terms) for water vapor (incompressible) with the Einstein notation yields:

$$\frac{\partial \overline{q}}{\partial t} + \overline{u}_j \frac{\partial \overline{q}}{\partial x_j} = -\frac{\partial (\overline{u'_j q'})}{\partial x_j}, \tag{1}$$

where $\overline{q}$ is the averaged specific humidity. Assuming horizontal homogeneity of the turbulent fluxes ($\frac{\partial}{\partial x}\overline{u'q'} = \frac{\partial}{\partial y}\overline{v'q'} = 0$), $\overline{w} = 0$, expressing the turbulent flux as latent heat flux ($\overline{w'q'} = LE/(\rho L_v)$) and taking into account that in a well-mixed boundary-layer $\overline{q}$ does not vary with height we can integrate Eq. (1) over height and get:

$$\underbrace{\frac{\partial \overline{q}}{\delta t}}_{I} = -\underbrace{\frac{\Delta\left(\frac{LE}{\rho L_v}\right)}{z_i}}_{II} - \underbrace{\overline{V}\frac{\partial \overline{q}}{\partial x}}_{III}, \tag{2}$$

where $\rho$ is the air density, $L_v$ is the heat of vaporization of water, $\Delta(LE/(\rho L_v))$ is the difference of latent heat flux between the top of the CBL and surface, $z_i$ is the CBL height, $\overline{V}$ is the average wind speed and $\partial/\partial x$ denotes differentiation along the average wind direction. The turbulent flux at the top of the CBL accounts for entrainment (including subsidence) and can be expressed as (Stull, 1988):

$$LE_{z_i} = \rho L_v w_e \Delta q = \rho L_v w_e \left[\overline{q}(z_i) - \overline{q}(z_i^+)\right], \tag{3}$$

with $\overline{q}(z_i)$ being the mean specific humidity in the CBL, $\overline{q}(z_i^+)$ is the specific humidity directly above the CBL and $w_e$ is the entrainment velocity. Without CBL height advection, the entrainment velocity is the difference between the local rate of a changing CBL height over time minus subsidence (Stull, 1988):

$$w_e = \frac{\partial z_i}{\partial t} - \overline{w}(z_i), \tag{4}$$

with $\overline{w}(z_i)$ as the vertical velocity at the height of the CBL. This results in the following expression for the difference in
latent heat flux between the surface ($LE_s$) and the top of the CBL:

$$\Delta\left(\frac{LE}{\rho L_v}\right) = \left[\overline{q}(z_i) - \overline{q}(z_i^+)\right] \cdot \left(\frac{\partial z_i}{\partial t} - \overline{w}(z_i)\right) - \frac{LE_s}{\rho_s L_v}, \tag{5}$$

where $\rho_s$ is the surface value of the air density. Equation (2) shows the the humidity tendency (Term I) with Term II representing the local (evapotranspiration) and term III the non-local contribution by horizontal advection. For ICON1 the terms of

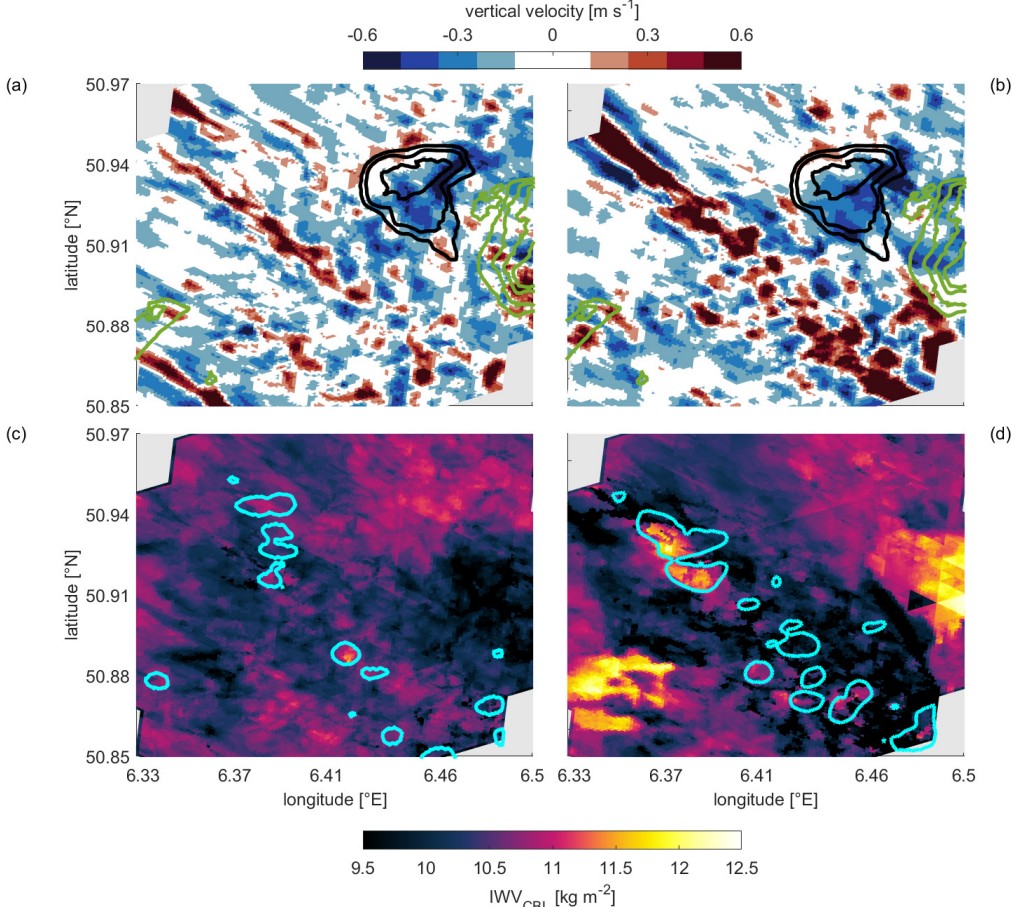

**Figure 7.** ICON-LEM vertically averaged vertical velocity (top) and integrated humidity (bottom) up to the CBL height of the ICON1 (a,c) and ICON2 (b,d) simulations. Contours in (a), (b) refer to the topography relative to JOYCE in $\mathrm{ma.s.l.}$ between -200 m to 0 m (green) and 0 m to 200 m (black) in 50 m steps. Contours in (c), (d) show areas with total column integrated cloud water values above 10 g m$^{-2}$. The results are averaged between 12–13 UTC.

Eq. (2) are calculated separately and averaged within the CBL for the domain showed in Fig. 5 between 10–11 UTC, where the CBL height increases from 500–770 m and still no clouds are present. During that time the specific humidity in the CBL increases by 0.62 g kg$^{-1}$ h$^{-1}$ on average. The contribution of the local term accounts for 0.21 g kg$^{-1}$ h$^{-1}$ and 0.29 g kg$^{-1}$ h$^{-1}$ is advected. This leaves a residual term of 0.12 g kg$^{-1}$ h$^{-1}$ indicating that the assumptions made for the budget equation are not valid, but it can be stated that in this simulation the humidity field is not entirely dominated by advection and the local source is of the same order of magnitude.

The sensitivity of slant path integrals of the water vapor field up to the CBL height to water vapor transport from the land surface can be evaluated from the local part of the humidity budget. Term II of Eq. (2) is calculated for a circular area with a radius corresponding to the projected CBL height of a 30° slant path and divided into sectors of 10°. Similar to the MWR measurements, the integrated water vapor is derived for a 30° slant path and 10° azimuth steps and integrated up to a height $h_{int}$ representing the maximum height of the slant path. At the normalized height of $h_{int}$ / CBL height = 1 the circle described by the slant path corresponds to the area where the local part of the humidity budget is computed. Figure 6 shows the correlation coefficient of the mean (10–11 UTC) 10° sector estimates of Term II of Eq. (2) and the slant path integrated water vapor at 30° elevation and 10° azimuth steps depending on the integration lengths. At short integration lengths no correlation between the integrated water vapor and the local source of humidity can be found. The correlation coefficient increases with height, reaches a maximum below the CBL height and decreases strongly above the CBL. This indicates that local sources of humidity at the surface can be detected by means of slant path integrated water vapor in a well-mixed boundary-layer when integrating up to the CBL height as performed in Sect. 3.2 with the MWR. Note that the values of the correlation coefficient would likely increase for cases with less advection.

Since it can be expected from the analysis of Eq. (2) that humidity transport from the surface is important for this day, changing the land use types is expected to have an influence on the development of the cloudy boundary-layer. In a second simulation (ICON2), the land use types are changed according to Fig. 5(b) (crop/grass to bare ground, bare ground to water, urban to forest, forest to crop/grass and water to urban). In this way, a significant reconstruction in the spatial distribution of the land use types is achieved without changing the scale of heterogeneity and keeping all occurring types. Also the partitioning of turbulent surface fluxes is largely affected by changing crop/grassland to bare soil, but for the whole simulation time the domain averaged sum of latent and sensible heat only differs by around 10 W m$^{-2}$ between ICON1 and ICON2. The maximum height above ground, where changing the land use types still has a significant influence on model parameters, is around 2.3–2.5 km, which is visible for example in the domain averaged specific humidity difference profile (not shown). Above this height the large scale forcings are more dominant, which are the same for both simulations. The highest difference occurs in the CBL, which is in agreement with Sühring and Raasch (2013) showing that heterogeneous surface patterns extend throughout the CBL for simulated turbulent heat fluxes. The length scale of land use variability seems to be large enough to cause these differences in the boundary-layer according to the blending-height concept (Mahrt, 2000). Also Shao et al. (2013) found an influence of land-surface heterogeneity well beyond the surface layer using LES.

In order to elaborate the details of different boundary-layer and cloud development, the spatial fields of height and time averaged vertical velocity and integrated humidity up to the CBL height (IWV$_{\mathrm{CBL}}$) are analyzed (Fig. 7). The averaging domain is the same as shown in Fig. 5 and the averaging time is between 12–13 UTC, which is the time range of the first cloud formation in the simulations. Poll et al. (2017) also performed large-eddy simulations of this day in a similar domain and showed the occurrence of clouds around this time in visible satellite data. They found cellular structures in the vertical velocity, which is also evident in Fig. 7(a). In addition, the wind is lifted by the hill and a downdraft above the hill can be seen. This was

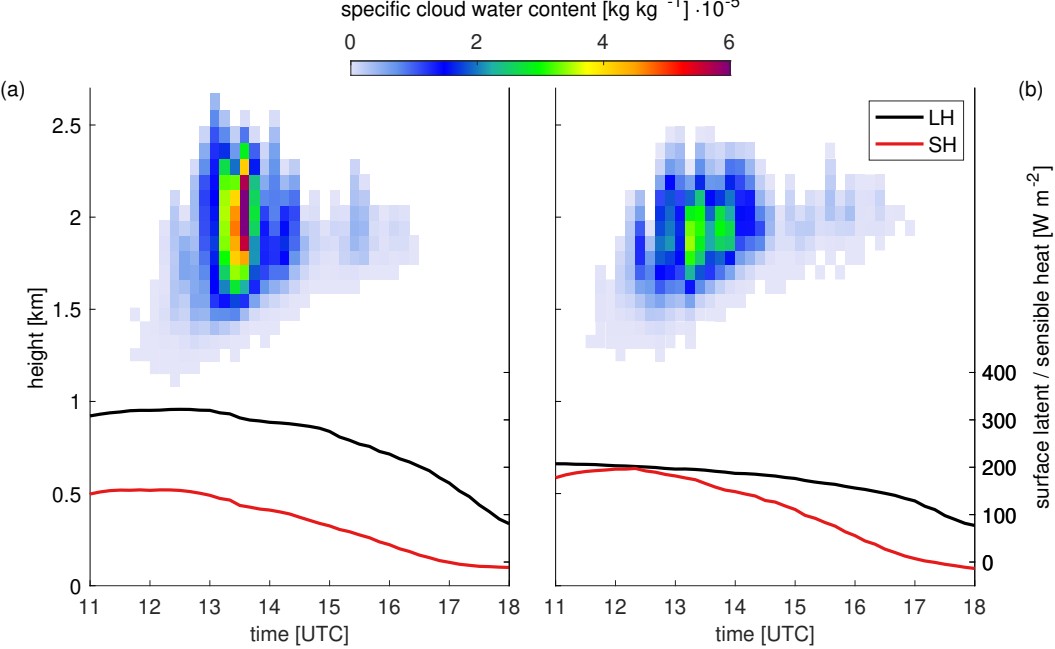

**Figure 8.** ICON-LEM specific cloud water content for the ICON1 (a) and ICON2 (b) simulation together with the surface fluxes of latent and sensible heat. The results are averaged for the domain shown in Fig. 5.

already discussed in Marke et al. (2018) and might explain parts of the negative scan deviations to the northeast, as discussed in Sect. 3.2, by a suppressed water vapor flux. Moreover the hill serves as a natural border and is impacting by channeling the updraft streak with associated water vapor transport and cloud formation going from northwest to southeast. The streaks are also visible in simulations using a larger domain, lower resolution, no topography and only bare ground (not shown), but

5    the position and strength is strongly altered by the topography and land use input. In the ICON2 simulation the differences in surface properties and the size of the heterogeneous land use patches intensifies the vertical velocity streak structure, leading to a higher water vapor transport from the surrounding area into the updraft region and an earlier cloud formation. The water bodies introduced in the second simulation show higher $IWV_{CBL}$ values (Fig. 7(d)), but sensible heat flux and CBL height are too low for clouds to form. The mean cloud cover of 8.55% in ICON1 compared to 10.55% in ICON2 is closer to the observed

10    maximum cloud cover of 6% determined by a total sky imager at JOYCE on this day.

Less vegetated areas and hence a lower roughness length in ICON2 also lead to an increase in the mean wind speed of $0.42\ \mathrm{m\,s^{-1}}$ at approximately 200 m above ground. With higher wind speeds and a higher fraction of bare ground the domain averaged sensible heat flux (between 11–18 UTC) in ICON2 is increased by $28.72\ \mathrm{W\,m^{-2}}$ and the CBL grows deeper (by about

15    30 m) especially in the southeastern part of the domain. On the other side the specific humidity in ICON1 is significantly larger in the CBL (Fig. 6) and clouds grow taller compared to the ICON2 simulation (Fig. 8), which is connected to an increased latent

heat flux by 86.04 W m$^{-2}$ in ICON1 due to more vegetated areas. Also the maximum integrated cloud water content of these clouds is 36.96 g m$^{-2}$ (ICON1) and only 5.61 g m$^{-2}$ in ICON2 because of the limited moisture supply. The drastic change in the land use data input for ICON2 therefore causes a shift in the partitioning between sensible and latent heat flux, which has strong implications for the development of convective clouds. Thus the long-term observed spatial water vapor deviations and

high-resolution LES conducted in this study underline the importance of further monitoring and modeling the local and small scale interactions between land use, topography, water vapor transport and the transition to clouds.

## 5   Conclusions

Exchange processes between the land surface and atmosphere are an important controlling factor in the water cycle. Long-term observational evidence of this interaction spanning scales of a few kilometers is still lacking. The scanning microwave

radiometer (MWR) at the Jülich ObservatorY for Cloud Evolution (JOYCE) proved to be suitable for detecting spatial IWV deviations for single scans, but also in a statistical sense. The atmospheric water vapor pattern can only partly be explained by the large-scale advection and is also attributed to the local transport of water vapor from the surface, especially during convective scenes. This is detected in the the long-term analysis of liquid water cloud free scans over six years of observations.

The comparison to the satellite-based MODIS near-infrared IWV product, as an independent observation, shows similar features of areas with pronounced positive and negative deviations around JOYCE. In a further step, these deviations can be related qualitatively to land surface properties by means of a land use classification. The classification is based on a remote sensing derived regional crop map and reveals that positive IWV deviations mainly originate over agricultural areas and open pit mines close to the measurement site, while urban and elevated forest areas show negative deviations. The main locations of

the maximum and minimum deviation in the MODIS and MWR measurements are in agreement, but seasonal effects related to the crop development stages are only visible in MWR observations.

In a comprehensive case study, large-eddy simulations with the high resolution ICON-LEM model were carried out to further assess the impact of the land surface on the development of the cloudy boundary-layer. While the control simulation is initiated

with a realistic land use input, the second simulation with modified land use types revealed changes in convective motions and cloud characteristics according to differences in surface fluxes. These findings suggest that ground-based remote sensing of water vapor supported by high resolution modeling can be valuable for studying the regional influence of heterogeneous land surfaces on the atmospheric water vapor and the connection between surface fluxes, water vapor and clouds.

*Author contributions.*  TM, SC, JHS and UL designed the experiments and processed the observational data. VS performed the ICON-LEM

simulations and TM prepared the manuscript with contributions from all co-authors.

*Competing interests.* The authors declare that they have no conflict of interest.

*Acknowledgements.* The authors would like to acknowledge the Transregional Collaborative Research Centre (TR32) "Patterns in Soil-Vegetation-Atmosphere Systems" funded by the German Science Foundation (DFG), which has continuously contributed to the instrumentation of JOYCE-CF and its maintenance as well as funding T. Marke. Further, the Humidity And Temperature PROfiler (HATPRO) used in this study have been funded by DFG infrastructural programs under the grant INST 216/681-1. The MODIS/Terra Total Precipitable Water Vapor 5-Min L2 Swath 1km dataset was acquired from the Level-1 and Atmosphere Archive & Distribution System (LAADS) Distributed Active Archive Center (DAAC), located in the Goddard Space Flight Center in Greenbelt, Maryland (https://ladsweb.nascom.nasa.gov/).

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
