# Peer review of "Detection of land surface induced atmospheric water vapor patterns"

_Atmospheric Chemistry and Physics, 2019_

## Referee Comment (RC1) · Wayne Angevine (Referee) · 20 May 2019

The discussion paper describes analysis of scanning radiometer measurements, finding persistent patterns in several years of data, and attempting to attribute those patterns to land use. It is an interesting dataset and interpretation, and should be published after some clarifications.

General comments: 1. The sort of patterns described here are, according to the literature, strongly wind-speed dependent. As far as I can tell, the wind speeds are not shown here. Does the strength of the patterns depend on the wind speed? There should be plenty of samples to support some binning of the data. See (and reference) the literature on "blending height."

[Figure]

2. A related point: Mesoscale circulations are mentioned in the introduction. They turn out to be rare in real data, mostly because of the wind speed dependence mentioned above. This comes up again in the discussion of fig.6, see specific comment below. Please consider how to make this point more carefully.

3. The information content of radiometer measurements is usually rather small (at most a few pieces of information in a beam). When the humidity profile is "integrated up to a scaling height of 2.5 km," (p.7 line 2), how many pieces of information are actually included in the integral? How do you know where the height of 2.5 km is in the slant path profile?

4. Why were the changed land use types chosen in the ICON2 simulation? It is not clear to me that the simulation with these choices really clarifies the issues in question. For example, if the hypothesis is that the mined areas emit more water vapor than other land use types, changing them to water bodies is not necessarily the best way to test that. Bare water doesn't emit water vapor very efficiently compared to some crops, for example. Can you explain or justify better why you made these particular choices?

Specific comments: 1. p.1 line 15: "Compartments" is maybe not the best word here. "Interactions between the land surface and the atmospheric boundary layer..." would be better.

2. p.13 line 4-5: I am not confident that there is really a secondary circulation here, or a roll structure. The quasi-linear features are present in both simulations, just a bit more clear in ICON2. If there is a roll structure, it's not necessarily related to the land use patches, rolls occur frequently even over homogeneous surfaces.

---

## Referee Comment (RC2) · Anonymous Referee #2 · 30 May 2019

**Atmos. Chem. Phys. Discuss., Manuscript Review**

**Manuscript number: ACP-2019-322**
**Reviewer: 2**

**Author(s):**
Tobias Marke, Ulrich Löhnert, Vera Schemann, and Susanne Crewell

**Title:**
Detection of land surface induced atmospheric water vapor patterns

**General Comments**
This paper aims to study "whether spatial water vapor distributions can be connected to land surface properties". It uses instruments from the ground-based remote sensing infra structure JOYCE and builds on a number of more technical papers, mostly produced as part of the TransRegio32 project, that for this particular site and instrument-set developed methodologies/algorithms for the radiometer retrievals, boundary layer height, etc., as well as inter-comparison studies, algorithms for landuse maps, virtual MWR scans from MODIS IWV measurements and the LES-LEM set-up.

As I understand it this paper wants to take advantage of all these tools that have been developed before to now focus on a process study into the influence of the surface to the amount of water vapour found in the atmosphere directly above these surfaces.

This in itself is an interesting objective and well worth publishing. However in my view the authors fail in the execution. There is too much attention on repeating all the technical work done before. It is not clear to me how some of the presented results contribute to answering the research questions (e.g. the inclusion of the MODIS data and the drastic landuse change in the LES). Furthermore, there is not much attention to the processes that make up the water vapour field in the atmosphere and once that would have been outlined continue with how to zoom in to contributions from the local land surface into the water vapour field as measured by the radiometer (a slant path over 4.3km in the horizontal reaching to 2.5km height).

This paper borders rejection in my view because it fails to orderly describe the processes at hand, come up with a good research strategy and presenting results that all lead to answering the research question. It would require a very major revision to fix this. In the following I will substantiate my statement with some specific points.

**Specific Issues**

1. Outline the main concepts:

   I miss a conceptual framework that described the water vapour concentration in the atmosphere, some notion of the scale of mixing in the horizontal and vertical and then connect these to the slant, integrated measurements of the MWR and land-use

map. The LES could serve as a means to distinguish processes by switching them on/off.

A simplified budget equation would be a good anchor for the analysis:

$$\frac{\Delta <q>}{\Delta t} = \frac{1}{\rho L_v}\frac{\Delta LE}{h} + u\frac{\Delta <q>}{\Delta x}$$

It shows the boundary layer humidity tendency as a function of a local source (evapotranspiration) and a non-local source (transport term) of humidity. h represents the boundary layer height.

Next step would be to connect some sense of scale to this equation on what sort of surface area the local part of the budget is sensitive to as the MWR beam transverses the atmosphere horizontally and vertically at the same time. The blending height concept might be useful here. What is the meaning of the part of the MWR path that is above the boundary layer in terms of its relation with the underlying surface?

The difficulty the MWR measurements is that transport will always play some role and at some height the connection with the surface directly below will be at least partly lost. This must be discussed and a clear approach must be sketched on how to zoom, as much as possible, into the local influence on the measurements. Large, synoptic scale transport if important will likely affect the whole scanning region equally, but can local transport (within the scanning beam) lead to a shift between scanning beam and underlying surface?

With the LES the two terms of the budget can be determined and evaluated separately. Also a run could be set-up with no local flux or no transport.

2. Section 3.1:

- To exclude advection you set an arbitrary limit to the humidity tendency (p7 lines18-25). Looking at the equation above it seems to make more sense to set a limit to the wind speed or look for synoptic weather patterns that go with low dq/dx (large scale high pressure situations). What is confusing is that 2 pages further on (p9 lines 11-12) you mention that you do an additional filtering on wind to rule out the transport? But supposedly you already filtered for this effect?
- What is the reason to lump all the cases with no clouds and little advection into one figure which combines situations from beginning to end of the growing season (crop-fields changing from bare soil to green to yellow) over a period of years (dry, wet conditions all mixed). Wouldn't it make more sense to present here the case you will study with the LES as well and off-set it against the multi-year picture and/or a situation in which advection does play an important role?
- Figure 2 is very busy. I suggest to separate the line plots from the surface plots. Also it Fig 2a is maybe easier to interpret if you plot it radially so it matches the scanning circle of Fig 1. In Fig1 you could indicate the 36 bins?

- Most of the text on pages 8-10 is on technical interpretation of the figures (e.g. you find out that the data filtering leads you to focus on anticyclonic weather types and that the WD compares well between methods); but there is very little on the relation between the MWR reading and the influence of the surface.

3. MODIS

The MODIS section reads as an intercomparison exercise of MWR measurements against constructed, virtual MWR scans from the MODIS data. They agree well, which is nice but the MODIS data doesn't provide additional insides on the research question. Fig3b without the MODIS line would have provided the same information relevant to the connection between MWR and the surface. I suggest to leave this section out and move Fig3b to section 3.1.

4. LES

- See final comment under point 1. The LES gives you the tool to investigate the relative importance of local influence vs transport under various conditions.
- I am not sure what the "inverted" landuse map teaches us. The MWR profile looks roughly the same (Fig 5) and moisture and cloud fields show similar patterns albeit with different intensity. Is it fair to conclude that the advection and topography are more important than landuse?

**Minor issues:**

1. Lines 11-12; Change "which was used in addition to investigate changes in surface fluxes and the water vapor and cloud field for an altered land use input." to "which was used to investigate changes in surface fluxes and the water vapor- and cloud fields for an altered land use input."

2. Line 10 (and corresponding reference in the literature list): "Guerau de Arellano (2008)" should be "Vilà-Guerau de Arellano (2008)"

---

## Referee Comment (RC3) · Anonymous Referee #3 · 17 Jun 2019

I have reviewed "Detection of land surface induced atmospheric water vapor patterns" by Marke et al. This study is focused on the analysis of observations and large-eddy simulations of water vapor over a relatively small area (approximately 12 by 13 km) near Jülich ObservatorY of Cloud Evolution (JOYCE). Overall, the paper provides an interesting look at features of the flow near the JOYCE site, and it will be relevant and of interest to scientists studying land-atmosphere interactions. While the presentation is generally clear and the reasoning is sound, there are some issues that should be addressed befor the manuscript is suitable for publication in the journal Atmospheric Chemistry and Physics.

Specific comments include: Page 1, Lines 15 and 22: Is "compartments" the correct word? Components might be better. The the word could be removed from Line 15. In

line 22 it could be replaced with "the surface" or something similar. Page 2, Lines 1-2: I am not familiar with the Transregional Collaborative Research Center 32 (TR32) Patterns in Soil-Vegetation-Atmosphere Systems, or how it might be important be in the context of this study? Some additional explanation would be helpful. Page 2, Lines 5-6: What is meant by high-resolution? Isn't LES by definition high-resolution compared to the scales of the flow? Page 3, Line 31: Do the years used in this study include a mix of relatively dry and wet years? Page 4, Line 1: The text states that linear interpolation is used for missing values. Is this treatment an issue if you are concerned about the details of the spatial pattern in the boundary layer? Page 4, Lines 18 and 19: The text states that the TKE dissipation rate is based on variance of the mean Doppler velocity. It would be helpful to have a few additional words about how the threshold is applied. Is it a threshold of dissipation, variance, or something different? Page 5, Line 14: It feels like there are some key aspects of the model configuration that are not covered in this section. For example, what data set is used for initial and boundary conditions or are they assumed to be periodic? These simulations could also be sensitive to the treatment of the land surface. It would be helpful to the reader to have some discussion of these important aspects. Figure 1: How does this domain compare to domain used by the ICON-LEM? Page 7, Lines 4-5: The text states "The main crop types between April and September are. . .". Isn't it important to differentiate between these various plant types that could have very different transpiration rates and how they are represented in the land-use/land-cover data set?? Page 7, Line 20: The sentence "The first and last scan of each sequence is neglected. . ." seems to contradict the previous one. Page 7, Line 26-34: The scan frequency of the MWR is likely much slower than the time scales of the turbulence. Thus, the data that is shown won't be able to resolve individual thermals. Is this a potential shortcoming, or does is the point to the need look at features with longer temporal scales? Page 7, Line 29: I do not entirely understand the sentence ". . .was found as mean value for the zenith MWR measurements. . .". It seems to indicate that the hourly mean value of IWV was determined based on 1 scan, but it seems that there should be additional scans in each hour-interval based on the

text in section 2.1. Page 10, Line 11: Some additional detail of how the virtual MWR scan is derived would be helpful as the details are not clear to me. Page 10, Lines 13-14: Why not use the boundary-layer height derived from the Doppler lidar? Page 11, Line 1: I believe that this is the first time that irrigation is mentioned in the manuscript. Is this a regular occurrence? Should it be mentioned earlier? Page 11, Lines 14-15: The text highlights differences in the observed and simulated boundary-layer depth. Why not just normalize the results using common boundary-layer scales? Maybe there isn't sufficient data? Figure 5: Could more tick marks be added to the horizontal axis of Figure 5? Maybe one every 45 or 90°? Page 12, Lines 6-9: The text states "Also a more dominate large scale humidity. . ..". This sentence argues for some additional discussion of the boundary-conditions use to drive the model. Figure 6: Over what depth was the vertical averaging applied? Page 13, Lines 4-6: Are the roles really a secondary circulation associated with the different amounts of moisture or are they simply a response to overall forcing in this particular case study? Back-of-the envelop calculations could be completed and compared to thresholds that have appeared in the literature. Page 14, Line 1: Is the change mentioned here associated with the intensity of the roles, or is it related to some other aspect of the flow? Page 14, Line 8: It would be clearer to use "larger" rather than "higher" in this sentence. Page 14 Line 10: Does the sum of the sensible and latent heat fluxes differ in the two simulations? It's hard to tell from Figure 7, and it could impact the interpretation of the results. Page 15, Line 3: Should "in" be "over" or some other word? Page 15, Line 6: Should "also" be added between "is" and "attributed"? Page 15, Line 16: "Are" should be "were". Page 15, Line 19: Is this really a mesoscale circulation, or is it smaller scale? Note that there was also a comment in section 3.3 regarding the changes in the winds and the nature of the changes in the boundary-layer flow. Would it be more accurate to simply say that there are changes in the boundary-layer flow structures?

---

## Author Comment (AC1) · 21 Aug 2019

Please find our detailed responses to the reviewers including the author's changes in manuscript and the revised manuscript in the supplement.

Please also note the supplement to this comment:
https://www.atmos-chem-phys-discuss.net/acp-2019-322/acp-2019-322-AC1-supplement.zip

---

## Author Response (AR1)

The authors would like to thank all three reviewers for their constructive and helpful comments. We are certain that by addressing and clarifying the issues mentioned by the reviewers, the quality of the revised manuscript has improved significantly. In the following we provide a detailed response to the **comments of all reviewers** together with the *changes in manuscript and where they occur*.

**Reviewer #1: Wayne Angevine**

**General comments:**

10 **1. The sort of patterns described here are, according to the literature, strongly wind-speed dependent. As far as I can tell, the wind speeds are not shown here. Does the strength of the patterns depend on the wind speed? There should be plenty of samples to support some binning of the data. See (and reference) the literature on "blending height."**

15 Beside looking for IWV pattern changes related to the wind direction, also the wind speed was taken into account, but no significant change in pattern strength during the course of the day was found. The median daily difference of maximum and minimum deviation from mean of the scan accounts for 1.1% for wind speeds below the median value of 5 m/s and 1.3% for wind speeds higher than the median.

Although, a change in the direction of the peaks due to wind speed and wind direction is evident, indicating a shift between the atmospheric water vapor pattern and underlying surface. This dependency is discussed in section 3.1 and an additional plot was added (Fig . 3):

25 *P. 9, l. 34: Separating all cases according to the low-level wind direction from the Doppler lidar, a directional dependence is found related to the wind speed. As an example, all MWR scans during northwesterly (270°-360°) winds are averaged separated by the median wind speed (5 m/s). Fig. 3 shows the highest positive deviations in the southwest direction for low wind speeds. For higher wind speeds this peak is shifted towards the southeast direction, indicating local transport and a*
30 *shift of the MWR scanning beam and the underlying surface. The peak around 70° is not significantly changing, possibly due to a wind shading effect of the hill to the northeast. This dependency can also be seen for other wind directions. To exclude this process and to better connect the spatial IWV deviations with the surrounding land use, the following analysis is restricted to cases with wind speeds below the median value.*

According to the blending height literature (reference Sühring and Raasch (2013) added) the strongest influence of the heterogeneous surface is assumed to be present up to the top of the convective boundary layer, which is considered here.

40 **2. A related point: Mesoscale circulations are mentioned in the introduction. They turn out to be rare in real data, mostly because of the wind speed dependence mentioned above. This comes up again in the discussion of fig.6, see specific comment below. Please consider how to make this point more carefully.**

45 In the introduction, mesoscale circulations are mentioned as an example for larger scale effects of land surface influence, but they are difficult to observe with the instrument setting at JOYCE. Therefore the focus is more on smaller scale features.

*P. 1, l. 18: On a more local scale the transport of energy and water vapor into the atmosphere can trigger the formation of shallow convective clouds and precipitation (e.g. Rabin et al., 1990; Avissar and Schmidt, 1998).*

**3. The information content of radiometer measurements is usually rather small (at most a few pieces of information in a beam). When the humidity profile is "integrated up to a scaling height of 2.5 km," (p.7 line 2), how many pieces of information are actually included in the integral? How do you know where the height of 2.5 km is in the slant path profile?**

The degrees of freedom for signal (DFS) in the lowest 2.5 km are usually 1-2 for MWR humidity retrievals and do not not show a significant increase for higher altitudes (Barrera-Verdejo et al. 2016). Therefore the highest information content can be found in the boundary layer. For the 90° (zenith) retrieval we identified 1.87 DFS (0.96 below 2.5 km) and for the 30° (slant path) retrieval

15  2.14 DFS (1.04 below 2.5 km). The height grid in the slant path profile is set in the forward modeling step of the retrieval derivation process in the same way as for the zenith retrieval.

*P. 4, l. 1: The degrees of freedom for signal (DFS) are usually between 1-2 for MWR humidity retrievals and the highest information content can be found in the boundary-layer. For the zenith*

20  *retrieval 1.87 DFS and for the 30° (slant path) retrieval 2.14 DFS are identified.*

**4. Why were the changed land use types chosen in the ICON2 simulation? It is not clear to me that the simulation with these choices really clarifies the issues in question. For example, if the hypothesis is that the mined areas emit more water vapor than other land use types, changing**

25  **them to water bodies is not necessarily the best way to test that. Bare water doesn't emit water vapor very efficiently compared to some crops, for example. Can you explain or justify better why you made these particular choices?**

The land use types are changed in a way that all types occur in both simulations without changing

30  the scale of heterogeneity. In this domain, the highest fraction is covered with crop/grass land. This is changed to bare ground in order to achieve a significant difference in the partitioning of the turbulent surface heat fluxes. The mining areas could potentially be a source for water vapor in the case of strong irrigation. Since we don't have access to the irrigation times and amount, this can not be verified. For the LES model to some extent this effect is simulated by changing the land use type

35  of the pit mines from bare soil to water.

*P. 13, l. 17: In a second simulation (ICON2), the land use types are changed according to Fig. 5(b) (crop/grass to bare ground, bare ground to water, urban to forest, forest to crop/grass and water to urban). In this way, a significant reconstruction in the spatial distribution of the land use types is*

40  *achieved without changing the scale of heterogeneity and keeping all occurring types. Also the partitioning of turbulent surface fluxes is largely affected by changing crop/grass land to bare soil, but for the whole simulation time the domain averaged sum of latent and sensible heat only differs by around 10 W/m² between ICON1 and ICON2.*

45  **Specific comments:**

**1. p.1 line 15: "Compartments" is maybe not the best word here. "Interactions between the land surface and the atmospheric boundary layer..." would be better.**

50  This part has been changed according to the suggestion:

*P. 1, l. 14: Interactions between the land surface and the atmospheric boundary layer can have significant influences on the regional weather and climate.*

**2. p.13 line 4-5: I am not confident that there is really a secondary circulation here, or a roll structure. The quasi-linear features are present in both simulations, just a bit more clear in ICON2. If there is a roll structure, it's not necessarily related to the land use patches, rolls occur frequently even over homogeneous surfaces.**

The authors agree that there is no clear evidence for a secondary circulation in this particular case and this wording is removed from the article. The features are even present in larger domain and lower resolution simulations without topography and bare ground as the only land use type. But changing the land use types for the second simulation shows a clear effect on the evolution of the boundary layer and cloud formation in terms of timing and characteristics. This can be attributed to the resulting local changes in surface fluxes, wind speed (lower roughness length for bare ground) and hence water vapor transport.

*P. 14, l. 20: The streaks are also visible in simulations using a larger domain, lower resolution, no topography and only bare ground (not shown), but the position and strength is strongly altered by the topography and land use input.*
*In the ICON2 simulation the differences in surface properties and the size of the heterogeneous land use patches intensifies the vertical velocity streak structure, leading to a higher water vapor transport from the surrounding area into the updraft region and an earlier cloud formation.*

**Reviewer #2: Anonymous Referee**

**1. Outline the main concepts**
**I miss a conceptual framework that described the water vapour concentration in the atmosphere, some notion of the scale of mixing in the horizontal and vertical and then connect these to the slant, integrated measurements of the MWR and land-use map.**
**The LES could serve as a means to distinguish processes by switching them on/off.**
**A simplified budget equation would be a good anchor for the analysis:**

$$\frac{\Delta <q>}{\Delta t} = \frac{1}{\rho L_v} \frac{\Delta LE}{h} + u \frac{\Delta <q>}{\Delta x}$$

**It shows the boundary layer humidity tendency as a function of a local source (evapotranspiration) and a non-local source (transport term) of humidity. h represents the boundary layer height.**
**With the LES the two terms of the budget can be determined and evaluated separately. Also a run could be set-up with no local flux or no transport.**

We agree, that the LES in combination with several sensitivity studies and an extended analysis could serve as a tool to disentangle different processes and investigate the contributions to the budget equation at least for a specific case study. But in our opinion this would be a separate study on its own. The focus of the presented manuscript is on the unique long-term analysis of the MWR scans and its use to investigate land surface induced patterns. The carefully selected cases show in average a pattern that can be explained by local land surface influences and can be connected to the land use types around JOYCE. The case study with the ICON-LEM is added to introduce the

potential of the model and test its capability to assess the impact of a changing land surface on the development of the cloudy boundary-layer. A more detailed study on the budget and to disentangle different processes would of course be interested and could be part of future research. Details are given below in the answers to the specific questions.

**Next step would be to connect some sense of scale to this equation on what sort of surface area the local part of the budget is sensitive to as the MWR beam transverses the atmosphere horizontally and vertically at the same time. The blending height concept might be useful here. What is the meaning of the part of the MWR path that is above the boundary layer in terms of its relation with the underlying surface?**

The case selection for the long-term analysis of MWR scans aim at minimizing larger scale effects like advection of dry/wet air masses. In this way the MWR beam would be sensitive to a surface area determined by the size of convective eddies. Here, a height of 2.5 km and therefore a horizontal area of 4.3 km for the scans with 30° elevation is chosen to include the convective boundary-layer and allow for a sufficient information content of the MWR. Within this height the degrees of freedom for signal (DFS) is still 1.04 compared to the overall 2.14 DFS for the slant path retrieval.

*P. 4, l. 1: The degrees of freedom for signal (DFS) are usually between 1-2 for MWR humidity retrievals and the highest information content can be found in the boundary-layer. For the zenith retrieval 1.87 DFS and for the 30° (slant path) retrieval 2.14 DFS are identified.*

A height of 2-2.5 km, where land use changes between ICON1 and ICON2 are causing differences in the simulations, can also be seen in the domain averaged relative difference of specific humidity (Fig. 6). This is also in agreement with the LES study of Shao et al. (2013), finding an influence of land-surface heterogeneity well beyond the surface layer.
Based on the case study simulation and the limited vertical resolution of the MWR, finding a general blending height is difficult.

*P. 14, l. 5: The maximum height above ground, where changing the land use types has still a significant influence on model parameters, is around 2.3-2.5 km, which is visible for example in the domain averaged specific humidity difference (ICON1-ICON2) profile (Fig. 6). Above this height the large scale forcings are more dominant, which are the same for both simulations. The highest difference occurs in the CBL, which is in agreement with Sühring and Raasch (2013) showing that heterogeneous surface patterns extend throughout the CBL for simulated turbulent heat fluxes. Also Shao et al. (2013) found an influence of land-surface heterogeneity well beyond the surface layer using LES.*

**The difficulty the MWR measurements is that transport will always play some role and at some height the connection with the surface directly below will be at least partly lost. This must be discussed and a clear approach must be sketched on how to zoom, as much as possible, into the local influence on the measurements. Large, synoptic scale transport if important will likely affect the whole scanning region equally, but can local transport (within the scanning beam) lead to a shift between scanning beam and underlying surface?**

When sorting the MWR scans with respect to wind direction and speed, a change in the water vapor pattern is visible indicating a shift between the scanning beam and fluxes emerging from the underlying surface corresponding to the wind direction for high wind speeds. This is now shown exemplary for the northwest wind direction (Fig. 3), which is also occurring in the case study. As a

consequence, an additional filter based on wind speed is applied, excluding all cases with a wind speed above 5 m/s for the further analysis.

*P. 9, l. 34: Separating all cases according to the low-level wind direction from the Doppler lidar, a directional dependence is found related to the wind speed. As an example, all MWR scans during northwesterly (270°-360°) winds are averaged separated by the median wind speed (5 m/s). Fig. 3 shows the highest positive deviations in the southwest direction for low wind speeds. For higher wind speeds this peak is shifted towards the southeast direction, indicating local transport and a shift of the MWR scanning beam and the underlying surface. The peak around 70° is not significantly changing, possibly due to a wind shading effect of the hill to the northeast. This dependency can also be seen for other wind directions. To exclude this process and to better connect the spatial IWV deviations with the surrounding land use, the following analysis is restricted to cases with wind speeds below the median value.*

**2. Section 3.1:**
**To exclude advection you set an arbitrary limit to the humidity tendency (p7 lines18-25). Looking at the equation above it seems to make more sense to set a limit to the wind speed or look for synoptic weather patterns that go with low dq/dx (large scale high pressure situations). What is confusing is that 2 pages further on (p9 lines 11-12) you mention that you do an additional filtering on wind to rule out the transport? But supposedly you already filtered for this effect?**

We agree that setting a limit to the wind speed is meaningful to exclude advection. As discussed in the previous comment, a filter for wind speed is added, excluding all cases with a wind speed above 5 m/s and the directional dependence is analyzed (Fig. 3).

**What is the reason to lump all the cases with no clouds and little advection into one figure which combines situations from beginning to end of the growing season (crop-fields changing from bare soil to green to yellow) over a period of years (dry, wet conditions all mixed). Wouldn't it make more sense to present here the case you will study with the LES as well and off-set it against the multi-year picture and/or a situation in which advection does play an important role?**

The general idea of the study was to identify situations when the surface shows the strongest effect on the moisture field. As a first proxy high advection situations were eliminated as during these the surface influence is low as for example shown by Steinke et al. (2019) by the reduction of the amplitude of the diurnal water vapor cycle. Different classifications (e.g. seasons) were applied, however, we did not succeed in identifying any significant changes in the patterns when sorting for these classes.

*P. 7, l. 21: This excludes overcast situations and large scale advection of moist or dry air as during these the surface influence is low as shown by Steinke et al. (2019) by the amplitude reduction of the diurnal water vapor cycle.*

In addition to the plot showing the scan deviation for situations with the same wind direction as in the case study (Fig. 3), also the multi-year picture (Fig. 4a) is shown and compared to the single case results (Fig. 4b) in section 3.2.

**Figure 2 is very busy. I suggest to separate the line plots from the surface plots. Also it Fig 2a is maybe easier to interpret if you plot it radially so it matches the scanning circle of Fig 1. In Fig1 you could indicate the 36 bins?**

Thank you for the suggestion. Figure 2 now only shows the line plots. The dissipation rate surface plot was removed since it doesn't give additional information compared to the CBL height. The scan deviation is now plotted radially and only for the period of interest during daytime. In addition azimuth angle bins are included in Fig. 1.

**Most of the text on pages 8-10 is on technical interpretation of the figures (e.g. you find out that the data filtering leads you to focus on anticyclonic weather types and that the WD compares well between methods); but there is very little on the relation between the MWR reading and the influence of the surface.**

As suggested in the previous comment, Fig. 2 is reduced by only showing the line plots and Fig. 3 (now Fig. 4) is replaced by a radial plot showing the long-term and case study scan deviations of MWR and MODIS in order to focus more on the relation between the MWR reading and the influence of the surface described in section 3.2.

**3. MODIS**
**The MODIS section reads as an intercomparison exercise of MWR measurements against constructed, virtual MWR scans from the MODIS data. They agree well, which is nice but the MODIS data doesn't provide additional insides on the research question. Fig3b without the MODIS line would have provided the same information relevant to the connection between MWR and the surface. I suggest to leave this section out and move Fig3b to section 3.1.**

The MODIS product is used as an additional and independent measurement for assessing the spatial water vapor distribution and to exclude a possible bias in the MWR data due to interference. The findings presented here could also be valuable for further studies using the MODIS products for assessing spatial IWV differences, which is especially valuable for larger areas.
This section is now shortened, Fig. 3a is removed and the MODIS result from Fig. 3b is compared to all MWR scans in section 3.2.

*P. 10, l. 15: For a comparison with an independent IWV measurement and to exclude that the patterns are influenced by interference, the MWR results are compared to the MODIS-NIR derived IWV around JOYCE. The findings presented here could also be valuable for further studies using the MODIS products for assessing spatial IWV differences, which is especially valuable for larger areas.*

**4. LES**
**See final comment under point 1. The LES gives you the tool to investigate the relative importance of local influence vs transport under various conditions.**

See discussion of final comment under point 1.

**I am not sure what the "inverted" landuse map teaches us. The MWR profile looks roughly the same (Fig 5) and moisture and cloud fields show similar patterns albeit with different intensity. Is it fair to conclude that the advection and topography are more important than landuse?**

The land use types are altered to achieve a strong change in the partitioning in surface heat fluxes and in a way that all types occur in both simulations.

We agree that in general advection and topography are more important, but here the intention was to identify the impact of the land use for low advection cases. Drawing conclusions on local water vapor patterns as done for the long-term MWR analysis is difficult on the basis of a single simulated day as it was visible from Fig. 5. Although a clear impact of the drastic land use change together with the topography is seen with respect to the evolution of the boundary layer and the formation of clouds. Therefore we focus now on the measured water vapor deviations by MWR and MODIS in section 3.2 and on the impact of the land use types on the development of the cloudy boundary-layer in section 4.

*P. 13, l. 9: In order to make a general statement whether the ICON-LEM is correctly representing the spatial water vapor distribution, several high resolution simulations would be needed. Here, the focus is on assessing the impact of different land use data as input for the model on boundary-layer development and cloud formation.*

**Minor issues:**

**1. Lines 11-12; Change "which was used in addition to investigate changes in surface fluxes and the water vapor and cloud field for an altered land use input." to "which was used to investigate changes in surface fluxes and the water vapor and cloud fields for an altered land use input."**

The text was modified incorporating the suggestion:

*P. 1, l. 10: In addition, high resolution large-eddy simulations (LES) are used to investigate changes in the water vapor and cloud fields for an altered land use input.*

**2. Line 10 (and corresponding reference in the literature list): "Guerau de Arellano (2008)" should be "Vilà-Guerau de Arellano (2008)"**

The reference was changed accordingly.

**Reviewer #3: Anonymous Referee**

**Page 1, Lines 15 and 22: Is "compartments" the correct word? Components might be better. The the word could be removed from Line 15. In line 22 it could be replaced with "the surface" or something similar.**

The word "compartments" has been removed:

*P. 1, l. 14: Interactions between the land surface and the atmospheric boundary layer can have significant influences on the regional weather and climate.*

*P. 1, l. 21: This requires assumptions near the surface boundaries, which strongly affects exchange processes.*

**Page 2, Lines 1-2: I am not familiar with the Transregional Collaborative Research Center 32 (TR32) Patterns in Soil-Vegetation-Atmosphere Systems, or how it might be important be in the context of this study? Some additional explanation would be helpful.**

This study was conducted within the framework of the Transregional Collaborative Research Center 32 (TR32). This statement together with the scope of TR32 is added to the text.

*P. 2, l. 1: The scope of TR32, as described in Simmer et al. (2015), is to improve the understanding and prediction capabilities of the spatiotemporal evolution of the terrestrial system across scales using measurement techniques and modeling platforms by integrating activities of several research groups.*

**Page 2, Lines 5-6: What is meant by high-resolution? Isn't LES by definition high-resolution compared to the scales of the flow?**

The authors agree that LES implies a high grid resolution to resolve turbulence and clouds. Although there are large differences in model resolution between different LES models and even within a single model, depending on the setup, ranging from 10-100 m (used in this study and referred to as high-resolution) to 100-200 m (usually used for larger domains).

**Page 3, Line 31: Do the years used in this study include a mix of relatively dry and wet years?**

The data used in this study covers five years. At this site the average year-to-year variability in terms of humidity is rather small, but still a good coverage of relatively dry and wet years was achieved in this study. As an exemplary measure, the mean zenith IWV taken around the selected scans for each year ranges from 15.2 kg/m² to 19.7 kg/m². The variability of the zenith IWV values for the different years (4.1-7.2 kg/m²) is in the range or higher than changes in the mean value.

*P. 8, l. 3: At JOYCE the average year-to-year variability in terms of humidity is rather small, but still a good coverage of relatively dry and wet years is achieved in this study. As an exemplary measure, the mean zenith IWV taken around the selected scans for each year ranges from 15.2 kg/m² to 19.7 kg/m². Therefore the variability of the zenith IWV values for the different years (4.1-7.2 kg/m²) is in the range or higher than changes in the mean value.*

**Page 4, Line 1: The text states that linear interpolation is used for missing values. Is this treatment an issue if you are concerned about the details of the spatial pattern in the boundary layer?**

Since these single scanning directions are not connected, which would create a larger gap, we are assuming a smooth transition at the gaps. Therefore excluding single azimuth directions due to interference and using linear interpolation is not causing an issue in this case.

*P. 4, l. 7: Since the excluded azimuth directions are not connected, no larger gap is apparent and a smooth transitions between the gaps can be assumed. Therefore the missing IWV values are filled using a linear interpolation.*

**Page 4, Lines 18 and 19: The text states that the TKE dissipation rate is based on variance of the mean Doppler velocity. It would be helpful to have a few additional words about how the threshold is applied. Is it a threshold of dissipation, variance, or something different?**

The threshold based approach for determining the CBL height using the TKE dissipation rate, as described in Manninen et al. (2018), finds the last range bin in each profile with significant turbulence in a bottom-up approach. This is now clarified in the text.

5 *P. 4, l. 25: The TKE dissipation rate is based on the variance of the observed mean Doppler velocity and allows for a threshold based estimation of the convective boundary-layer (CBL) height by determining the last range bin in each profile with significant turbulence in a bottom-up approach.*

10 **Page 5, Line 14: It feels like there are some key aspects of the model configuration that are not covered in this section. For example, what data set is used for initial and boundary conditions or are they assumed to be periodic? These simulations could also be sensitive to the treatment of the land surface. It would be helpful to the reader to have some discussion of these important aspects.**

The model is forced with ECMWF Integrated Forecasting System (IFS) data, which includes initial and lateral boundary conditions (so non-periodic).
As the IFS and the ICON model don't use an identical land surface model, we cannot exclude that the simulations are sensitive against the treatment of soil moisture and other land surface
20 components. But those sensitivities are the same for both simulations and a model simulation will always be sensitive to the setup - including domain size, resolution, forcing data, simulation time etc. But so far sensitivity studies implicate, that the results are rather robust despite small variations.

*P. 6, l. 2: As the IFS and the ICON model are not using an identical land surface model, a*
25 *sensitivity of the simulations against the treatment of soil moisture and other land surface components can not be excluded. But those sensitivities are the same for both simulations and sensitivity studies implicate, that the results are rather robust despite small variations.*

**Figure 1: How does this domain compare to domain used by the ICON-LEM?**

The domain of the land use type classification shown in Fig. 1 covers the main part of the ICON-LEM domain (10 km radius domain size) shown in Fig. 6 (now Fig. 7).

**Page 7, Lines 4-5: The text states "The main crop types between April and September are. . .".**
35 **Isn't it important to differentiate between these various plant types that could have very different transpiration rates and how they are represented in the land-use/land-cover data set??**

One has to note that the region is characterized by many small field with sizes of only a few 100 m
40 (see Fig. 1). Due to crop rotation the different crop types change from year to year and a rather random pattern of sugar beet and winter wheat fields next to each other can be found. Therefore, as no larger continuous areas of individual crop types occur, differences between these can not be resolved using the MWR.

45 *P. 7, l. 13: Due to this crop rotation and regarding the small field sizes in this domain, no further distinction in crop types is made.*

**Page 7, Line 20: The sentence "The first and last scan of each sequence is neglected. . ." seems to contradict the previous one.**
50

A sequence of at least three consecutive scans fulfilling the criteria are required to be included in the analysis. The beginning and ending of each sequence is neglected to ensure that they are not part of a transition from conditions violating the criteria.

*P. 7, l. 30: The first and last scan of each sequence is neglected to ensure that they are not part of a transition from conditions violating the criteria.*

**Page 7, Line 26-34: The scan frequency of the MWR is likely much slower than the time scales of the turbulence. Thus, the data that is shown won't be able to resolve individual thermals. Is this a potential shortcoming, or does is the point to the need look at features with longer temporal scales?**

Due to the scanning frequency the individual thermals, which are assumed to resemble the ensemble of thermals, transporting water vapor can not be detected with this method. Therefore we were aiming at detecting signals that are present in a composite of MWR scans as shown in section 3.2.

**Page 7, Line 29: I do not entirely understand the sentence ". . .was found as mean value for the zenith MWR measurements. . .". It seems to indicate that the hourly mean value of IWV was determined based on 1 scan, but it seems that there should be additional scans in each hour-interval based on the text in section 2.1.**

Here we only refer to the height, where the humidity profile drops below 1/e. This height was found by averaging the 1/e heights from all zenith measurements of the MWR 1 h around the scans, which have a temporal resolution of 1-2 seconds.

*P. 9, l. 1: This height, where the humidity profile drops below 1/e, was found by averaging the 1/e heights from all zenith measurements of the MWR within 1 h around each of the selected scans.*

**Page 10, Line 11: Some additional detail of how the virtual MWR scan is derived would be helpful as the details are not clear to me.**

The MODIS derived IWV is converted into a absolute humidity profile for each pixel assuming a linear decrease (by 20%) in the boundary-layer and an exponential decrease above. The mean CBL height is determined by the Doppler lidar and the 1/e height for the exponential decrease is derived from the MWR humidity profiles 1 h around the corresponding overpasses. This section is now updated with more details.

*P. 11, l. 1: Therefore the total IWV is distributed to an absolute humidity profile for each MODIS pixel assuming a linear decrease by 20% in the CBL and an exponential decrease above similar to Schween et al. (2011). The mean CBL height is determined from the Doppler lidar based boundary-layer classification Manninen et al. (2018) around 1 h of each overpass.*

**Page 10, Lines 13- 14: Why not use the boundary-layer height derived from the Doppler lidar?**

The boundary-layer classification is based on the Doppler lidar as described in section 2.2. For clarification, this is now repeated in this part.

*P. 11, l. 3: The mean CBL height is determined from the Doppler lidar based boundary-layer classification Manninen et al. (2018) around 1 h of each overpass.*

**Page 11, Line 1: I believe that this is the first time that irrigation is mentioned in the manuscript. Is this a regular occurrence? Should it be mentioned earlier?**

Irrigation in the pit mines, in contrast to no irrigation for the crop fields, is a potential source for water vapor, but there is no information available on the timing and amount of irrigation in this area, which makes it difficult to quantify. Therefore it is only briefly mentioned.

**Page 11, Lines 14-15: The text highlights differences in the observed and simulated boundary-layer depth. Why not just normalize the results using common boundary-layer scales? Maybe there isn't sufficient data?**

The exact height of the boundary layer is difficult to compare between one model simulation and observations due to different methods used to define it and using common boundary-layer scales would indeed require more data.

**Figure 5: Could more tick marks be added to the horizontal axis of Figure 5? Maybe one every 45 or 90°?**

The water vapor deviation derived from the scans are now represented as polar plots to allow for a better comparison with the surrounding land use and topography shown in Fig 1.

**Page 12, Lines 6-9: The text states "Also a more dominate large scale humidity. . ..". This sentence argues for some additional discussion of the boundary-conditions use to drive the model.**

This sentence was removed since drawing conclusions on local water vapor patterns as done for the long-term MWR analysis is difficult on the basis of a single simulated day as it was visible from Fig. 5. Although a clear impact of the drastic land use change together with the topography is seen with respect to the evolution of the boundary layer and the formation of clouds. Therefore we focus now on the measured water vapor deviations by MWR and MODIS in section 3.2 and on the impact of the land use types on the development of the cloudy boundary-layer in section 4.

*P. 13, l. 9: In order to make a general statement whether the ICON-LEM is correctly representing the spatial water vapor distribution, several high resolution simulations would be needed. Here, the focus is on assessing the impact of different land use data as input for the model on boundary-layer development and cloud formation.*

**Figure 6: Over what depth was the vertical averaging applied?**

The vertical averaging was applied to the whole column. This is now changed to the lowest 2.5 km to show that the structures emerge from the boundary layer. This statement is also added to the text:

*P. 14, l. 12: In order to elaborate the details of different boundary-layer and cloud development, the spatial fields of height and time averaged vertical velocity and integrated humidity up to 2.5 km (IWV_2.5) are analyzed (Fig. 7).*

and the caption of Fig. 6 (now Fig. 7):

*ICON-LEM vertically averaged vertical velocity (top) and integrated humidity (bottom) up to 2.5 km of the ICON1 (a,c) and ICON2 (b,d) simulations.*

**Page 13, Lines 4-6: Are the roles really a secondary circulation associated with the different amounts of moisture or are they simply a response to overall forcing in this particular case study? Back-of-the envelop calculations could be completed and compared to thresholds that have appeared in the literature.**

The authors agree that there is no clear evidence for a secondary circulation in this particular case and this wording is removed from the article. The features are even present in larger domain and lower resolution simulations without topography and bare ground as the only land use type. But changing the land use types for the second simulation shows a clear effect on the evolution of the boundary layer and cloud formation in terms of timing and characteristics. This can be attributed to the resulting local changes in surface fluxes, wind speed (lower roughness length for bare ground) and hence water vapor transport.

*P. 14, l. 20: The streaks are also visible in simulations using a larger domain, lower resolution, no topography and only bare ground (not shown), but the position and strength is strongly altered by the topography and land use input.*
*In the ICON2 simulation the differences in surface properties and the size of the heterogeneous land use patches intensifies the vertical velocity streak structure, leading to a higher water vapor transport from the surrounding area into the updraft region and an earlier cloud formation.*

**Page 14, Line 1: Is the change mentioned here associated with the intensity of the roles, or is it related to some other aspect of the flow?**

The change mentioned here refers to the enhanced vertical water vapor transport in the southeast part of the domain, where stronger updrafts and an increased mixing layer height can be observed compared to the reference simulation.

*P. 14, l. 31: With higher wind speeds and a higher fraction of bare ground the domain averaged sensible heat flux (between 11-18 UTC) in ICON2 is increased by 28.72 W/m² and the CBL grows deeper (by about 30 m) especially in the southeastern part of the domain.*

**Page 14, Line 8: It would be clearer to use "larger" rather than "higher" in this sentence.**

"higher" is replaced by "larger"

*P. 14, l. 33: On the other side the specific humidity in ICON1 is significantly larger in the CBL (Fig. 6) and clouds grow taller compared to the ICON2 simulation (Fig. 8), which is connected to an increased latent heat flux by 86.04 W/m² in ICON1 due to more vegetated areas.*

**Page 14 Line 10: Does the sum of the sensible and latent heat fluxes differ in the two simulations? It's hard to tell from Figure 7, and it could impact the interpretation of the results.**

For a domain average between 11-18 UTC, the sum of latent and sensible heat is 57.85 W/m² higher in ICON1 (86.04 W/m² higher latent heat flux in ICON1 and 28.72 W/m² higher sensible heat flux in ICON2). For the whole simulations the fluxes only differ by around 10 W/m².

*P. 13, l. 3: Also the partitioning of turbulent surface fluxes is largely affected by changing crop/grass land to bare soil, but for the whole simulation time the domain averaged sum of latent and sensible heat only differs by around 10 W/m² between ICON1 and ICON2.*

**Page 15, Line 3: Should "in" be "over" or some other word?**

"in" is replaced by "spanning"

*P. 16, l. 2: Long-term observational evidence of this interaction spanning scales of a few kilometers is still lacking.*

**Page 15, Line 6: Should "also" be added between "is" and "attributed"?**

The word "also" was added to the text.

*P. 16, l. 5: The atmospheric water vapor pattern can only partly be explained by the large-scale driven advection and is also attributed to the local transport of water vapor from the surface, especially during convective scenes.*

**Page 15, Line 16: "Are" should be "were".**

"are" is replaced by "were"

*P. 17, l. 1: In a comprehensive case study, large-eddy simulations by the high resolution ICON-LEM model were carried out to further assess the impact of the land surface on the development of the cloudy boundary-layer.*

**Page 15, Line 19: Is this really a mesoscale circulation, or is it smaller scale? Note that there was also a comment in section 3.3 regarding the changes in the winds and the nature of the changes in the boundary-layer flow. Would it be more accurate to simply say that there are changes in the boundary-layer flow structures?**

Please see the discussion to the previous comment on Page 13, Lines 4-6.

[revised manuscript text omitted]

---

## Referee Report (RR1)

**Atmos. Chem. Phys. Discuss., Manuscript Re-Review**

**Manuscript number: ACP-2019-322**
**Reviewer: 2**

**Author(s):**
Tobias Marke, Ulrich Löhnert, Vera Schemann, and Susanne Crewell

**Title:**
Detection of land surface induced atmospheric water vapor patterns

**General Comments**

I am not satisfied with the response to my review. A fair bit of cherry picking in the results is used to counter the issues I raised. Put together, these responses don't add up.

The goal is to look for the influence of the underlying landuse on observed water vapor in MWR scans. The scans are diagonal over 4.3km and reach upto 2.5km height above the surface. The authors also use a LES and MODIS images to evaluate the situation. Main comments on their approach were to do with

1) the question what the MWR ray actually sees of the underlying surface given the variable height reaching way above the boundary layer, weather conditions and the varying footprint and blending height and

2) what part is the water vapor signal is local and what part is advected. I proposed to study this using a simplified budget equation and use the LES to evaluate that.

All suggestions to bring MWR, satellite remote sensing product and LES closer together and more focused on the research question are ignored for various and contradictory reasons:

ISSUE: Missing framework (budget equation) to guide the research and separate local vs non-local contributions from the humidity field observed with the MWR. The LES could be used to distinguish the relative contribution of sources as well.

REPLY: "... in our opinion this would be a separate study on its own. The focus of the presented manuscript is on the unique long-term analysis of the MWR scans and its use to investigate land surface induced patterns."

RE-REPLY: I don't agree. This paper lacks a good framework to analyze the data for the goal as defined originally (link MWR to local landuse). In addition, it doesn't make sense to not use tools that you have available that are so valuable in answering your research question (use LES to separate local vs non-local contributions).
I now read that you define a second goal which is to highlight the unique long-term dataset available. This could also be an approach in analyzing the MWR dataset, but requires a redesign of the paper.

ISSUE: What is felt of the local surface at 2.5km height in light of the footprint of the measurements and the blending height concept?

REPLY (new txt in ms): "... The maximum height above ground, where changing the land use types has still a significant influence on model parameters, is around 2.3-2.5 km, which is visible for example in the domain averaged specific humidity difference (ICON1-ICON2) profile (Fig. 6).."

RE-REPLY: I am not convinced that the boundary layers in these particular LES runs are representative for the long term MWR dataset that you are analyzing. The LES runs are for the most extreme possible landuse signal (inversion of landuse), for one day in the hottest part of the year (end of July). Only then you see a signal upto 2.2km (not higher) in the humidity difference plot. But this is not the typical boundary layer for your long term dataset. Local conditions are not felt beyond the boundary layer height and the boundary layer height typically doesn't exceed ~1km as shown in Fig2a; so when integrating MWR signals upto 2.5m a large part of the signal will not be related to the underlying surface.

ISSUE: Why lumping all data over a season, as it is the seasonal change over time that will provide a strong change in moisture at the surface (crops growing, rain events, etc)?

REPLY: "The general idea of the study was to identify situations when the surface shows the strongest effect on the moisture field." .... "Different classifications (e.g. seasons) were applied, however, we did not succeed in identifying any significant changes in the patterns when sorting for these classes."

RE-REPLY: The fact that the widely spread agricultural fields in the area which undergo a transition from bare soil to highly evaporating green surfaces to dry ripened vegetation and back to bare soil doesn't leave a noticeable trace in the MWR signals seems to indicate that the MWR indeed doesn't see much of the surface when integrating the signals upto 2.5km

ISSUE: The LES results suggest that topography and advection are dominant over landuse in humidity signals

REPLY: " We agree that in general advection and topography are more important, but here the intention was to identify the impact of the land use for low advection cases. Drawing conclusions on local water vapor patterns as done for the long-term MWR analysis is difficult on the basis of a single simulated day as it was visible from Fig. 5. ...."

RE-REPLY: You are saying there is too much advection in the LES? I see that the wind speeds was 3m/s, well within your non-advection criterion. Also, you have full control over the LES, how can you say there is too much wind? You say one day is not enough to draw conclusions, so include more days. The main message I get, again, is that the local influence is relatively small.

ISSUE: to what extent do the MODIS images help to evaluate the identification of the landuse in the water vapour signals
REPLY: "The findings presented here could also be valuable for further studies using the MODIS products for assessing spatial IWV differences, which is especially valuable for larger areas."

RE-REPLY: Make up your mind about the goal is of this study. The MODIS part provides a nice inter-comparison but it is not related to the main research question.

The feeling I get when I read the paper is the following: from the start the hypothesis was that landuse leaves an imprint on the MWR signals. In my opinion, you bend the interpretation of your results too much to corroborate this hypothesis. Whereas your results indicate that the influence of landuse on the MWR signal is limited given the long MWR path that extends well beyond the boundary layer. Filtering the data for non-advection conditions doesn't help. Filtering the MWR data for conditions with known differences in landuse and soil moisture doesn't give a landuse signal. In an LES run with an extreme change in landuse advection and topography are dominant over landuse.

In all, I stick with the same verdict as the initial review: "This paper borders rejection in my view because it fails to orderly describe the processes at hand, come up with a good research strategy and presenting results that all lead to answering the research question."

The authors should either change the scope of their paper, i.e. change the research question or change the way they analyze the data. Right now they keep with the landuse imprint on the MWR data but all comments that question their approach are dismissed.

---

## Author Response (AR2)

**Atmos. Chem. Phys. Discuss., Manuscript Re-Review**

**Manuscript number: ACP-2019-322**
**Reviewer: 2**

**Author(s):**
Tobias Marke, Ulrich Löhnert, Vera Schemann, and Susanne Crewell

**Title:**
Detection of land surface induced atmospheric water vapor patterns

**General Comments**

I am not satisfied with the response to my review. A fair bit of cherry picking in the results is used to counter the issues I raised. Put together, these responses don't add up.

The goal is to look for the influence of the underlying landuse on observed water vapor in MWR scans. The scans are diagonal over 4.3km and reach upto 2.5km height above the surface. The authors also use a LES and MODIS images to evaluate the situation. Main comments on their approach were to do with

1) the question what the MWR ray actually sees of the underlying surface given the variable height reaching way above the boundary layer, weather conditions and the varying footprint and blending height and
2) what part is the water vapor signal is local and what part is advected. I proposed to study this using a simplified budget equation and use the LES to evaluate that.

All suggestions to bring MWR, satellite remote sensing product and LES closer together and more focused on the research question are ignored for various and contradictory reasons:

ISSUE: Missing framework (budget equation) to guide the research and separate local vs non-local contributions from the humidity field observed with the MWR. The LES could be used to distinguish the relative contribution of sources as well.

REPLY: "... in our opinion this would be a separate study on its own. The focus of the presented manuscript is on the unique long-term analysis of the MWR scans and its use to investigate land surface induced patterns."

RE-REPLY: I don't agree. This paper lacks a good framework to analyze the data for the goal as defined originally (link MWR to local landuse). In addition, it doesn't make sense to not use tools that you have available that are so valuable in answering your research question (use LES to separate local vs non-local contributions).
I now read that you define a second goal which is to highlight the unique long-term dataset available. This could also be an approach in analyzing the MWR dataset, but requires a redesign of the paper.

**AUTHORS RESPONSE:** The LES simulation is now being used to separate local and non-local contributions as suggested using the simplified humidity budget equation (section 4). Estimates of the different terms are derived during the development of the convective boundary-layer, showing that advection is not the dominant factor for the increase in humidity on that day. Even though the budget is not closed, indicating that some of the assumptions made are not valid, it shows that changes made concerning the land use types have a significant contribution to the humidity tendency.

ISSUE: What is felt of the local surface at 2.5km height in light of the footprint of the measurements and the blending height concept?

REPLY (new txt in ms): "... The maximum height above ground, where changing the land use types has still a significant influence on model parameters, is around 2.3-2.5 km, which is visible for example in the domain averaged specific humidity difference (ICON1-ICON2) profile (Fig. 6).."

RE-REPLY: I am not convinced that the boundary layers in these particular LES runs are representative for the long term MWR dataset that you are analyzing. The LES runs are for the most extreme possible landuse signal (inversion of landuse), for one day in the hottest part of the year (end of July). Only then you see a signal upto 2.2km (not higher) in the humidity difference plot. But this is not the typical boundary layer for your long term dataset. Local conditions are not felt beyond the boundary layer height and the boundary layer height typically doesn't exceed ~1km as shown in Fig2a; so when integrating MWR signals upto 2.5m a large part of the signal will not be related to the underlying surface.

**AUTHORS RESPONSE:** The authors agree that integrating the MWR signals up to 2.5 km would include parts that are not related to the surface since the boundary-layer height is usually lower. Therefore, the MWR signal is now integrated only up to the boundary-layer height estimated by the Doppler lidar. Please notice that this reduces the number of scans included in the analysis since measurements from both instruments need to be available.
In this way also seasonal effects related the growing stage of the dominant crop types are evident when dividing the data set into April-June and July-September cases (Fig. 3). The single case study analysis showed that the late July case still shows similar features compared to the April-June period, where positive IWV deviations are found in the direction of a crop dominated area (Fig. 4).
In addition, the local term of the humidity budget equation is used to demonstrate that the highest correlation of the surface fluxes and the slant path integrals of the water vapor can be found when integrating up to about the boundary-layer height (Fig. 6).

ISSUE: Why lumping all data over a season, as it is the seasonal change over time that will provide a strong change in moisture at the surface (crops growing, rain events, etc)?

REPLY: "The general idea of the study was to identify situations when the surface shows the strongest effect on the moisture field." .... "Different classifications (e.g. seasons) were applied, however, we did not succeed in identifying any significant changes in the patterns when sorting for these classes."

RE-REPLY: The fact that the widely spread agricultural fields in the area which undergo a transition from bare soil to highly evaporating green surfaces to dry ripened vegetation and back to bare soil doesn't leave a noticeable trace in the MWR signals seems to indicate that the MWR indeed doesn't see much of the surface when integrating the signals upto 2.5km

**AUTHORS RESPONSE:** With the change of integrating the MWR signals only up to the boundary-layer height estimated from the Doppler lidar (see comment above), seasonal effects can be detected and attributed to the surrounding land use (Fig. 3).

ISSUE: The LES results suggest that topography and advection are dominant over landuse in humidity signals

REPLY: " We agree that in general advection and topography are more important, but here the intention was to identify the impact of the land use for low advection cases. Drawing conclusions on local water vapor patterns as done for the long-term MWR analysis is difficult on the basis of a single simulated day as it was visible from Fig. 5. "

RE-REPLY: You are saying there is too much advection in the LES? I see that the wind speeds was 3m/s, well within your non-advection criterion. Also, you have full control over the LES, how can you say there is too much wind? You say one day is not enough to draw conclusions, so include more days. The main message I get, again, is that the local influence is relatively small.

**AUTHORS RESPONSE:** With the analysis of the humidity budget equation (section 4), it was found that the advection term is not dominant and local sourced contribute to almost the same amount to changes in the humidity field. In order get a significant change in surface fluxes and therefore in the local influence, the drastic change in land use types was chosen.

ISSUE: to what extent do the MODIS images help to evaluate the identification of the landuse in the water vapour signals

REPLY: "The findings presented here could also be valuable for further studies using the MODIS products for assessing spatial IWV differences, which is especially valuable for larger areas."

RE-REPLY: Make up your mind about the goal is of this study. The MODIS part provides a nice inter-comparison but it is not related to the main research question.

**AUTHORS RESPONSE:** The goal of this study is to evaluate whether the heterogeneity of land use patches surrounding the measurement site creates a spatial atmospheric water vapor pattern that can be measured. The MODIS part is thought of an independent measurement to determine if the land use patches are large enough to produce a pattern in the MODIS measurements with a rather coarse spatial resolution of 1 km. This is verified by the general agreement of the direction of positive IWV deviations between MWR and MODIS in the direction of crop/grass lands, pit mines and the negative deviation of the forested hill, although a seasonal signal is not evident (Fig. 3).

The feeling I get when I read the paper is the following: from the start the hypothesis was that landuse leaves an imprint on the MWR signals. In my opinion, you bend the interpretation of your results too much to corroborate this hypothesis. Whereas your results indicate that the influence of landuse on the MWR signal is limited given the long MWR path that extends well beyond the boundary layer. Filtering the data for non-advection conditions doesn't help. Filtering the MWR data for conditions with known differences in landuse and soil moisture doesn't give a landuse signal. In an LES run with an extreme change in landuse advection and topography are dominant over landuse.

In all, I stick with the same verdict as the initial review: "This paper borders rejection in my view because it fails to orderly describe the processes at hand, come up with a good research strategy and presenting results that all lead to answering the research question."

The authors should either change the scope of their paper, i.e. change the research question or change the way they analyze the data. Right now they keep with the landuse imprint on the MWR data but all comments that question their approach are dismissed.

**LIST OF RELEVANT CHANGES:**

In Figure 1 crop and grass land are now separated, to show the grass land area to the southwest where a positive IWV deviation was found.

In section 3 the slant path IWV of the MWR is now integrated up the top of the convective boundary-layer (determined by the Doppler lidar) instead of up to 2.5 km, which decreases the number of scans included in the analysis since measurements of both instruments are needed.

The analysis of the long-term IWV deviations from the MWR and MODIS is performed for different seasons (April-June, July-September) to demonstrate differences that can be attributed to the land use (growing stage of crops). This is shown in Fig. 3 and the case study results of MWR and MODIS are shown in Fig. 4.

In section 4 the humidity budget equation is used to separate local and non-local contributions to the humidity field as suggested by the reviewer. It could be demonstrated that the local term is contributing to the change in humidity so that a change in land use types would have a significant influence.
In addition, instead of showing the humidity difference between both simulations, a correlation analysis shows the connection of the slant path IWV to the local term of the budget for different integration lengths (Fig. 6).

The vertical velocity and IWV in Fig. 7 are now also only considered up the top of the convective boundary-layer.

Jan H. Schween contributed to the preparation of the revised manuscript and was added as co-author.

[revised manuscript text omitted]

---

## Author Response (AR3)

[revised manuscript text omitted]

**Response to Editor**

**p. 7, l. 27:** In order to detect significant changes in IWV with time that can be attributed to large scale humidity advection, a threshold above the instrument sensitivity is chosen. As the results suggest, the instrument still fulfills the requirements to detect spatial variability in the humidity field, especially in the long-term analysis.

**p. 16, l. 26:** A reference (Mahrt, 2000) and a statement was added to mention that the local length scale of land use variability around JOYCE seems to be large enough to affect the boundary layer based on the blending height concept.